

# MOPSMAP v0.9: A versatile tool for modeling of aerosol optical properties

Josef Gasteiger[1] and Matthias Wiegner[2]

[1]Faculty of Physics, University of Vienna, Vienna, Austria
[2]Meteorologisches Institut, Ludwig-Maximilians-Universität, München, Germany

*Correspondence to:* Josef Gasteiger (josef.gasteiger@univie.ac.at)

**Abstract.**

The spatiotemporal distribution and characterization of aerosol particles are usually determined by remote sensing and optical in-situ measurements. These measurements are indirect with respect to microphysical properties and thus inversion techniques are required to determine the aerosol microphysics. Scattering theory provides the link between microphysical and

optical properties; it is not only needed for such inversions but also for radiative budget calculations and climate modeling. However, optical modeling can be very time consuming, in particular if non-spherical particles or complex ensembles are involved.

In this paper we present the MOPSMAP package (modeled optical properties of ensembles of aerosol particles) which is computationally fast for optical modeling even in case of complex aerosols. The package consists of a data set of pre-calculated

optical properties of single aerosol particles, a Fortran program to calculate the properties of user-defined aerosol ensembles, and a user-friendly web interface for online calculations. Spheres, spheroids, and a small set of irregular particle shapes are considered over a wide range of sizes and refractive indices. MOPSMAP provides the fundamental optical properties assuming random particle orientation, including the scattering matrix for the selected wavelengths. Moreover, the output includes tables of frequently used properties such as the single scattering albedo, the asymmetry parameter or the lidar ratio. To demonstrate

the wide range of possible MOPSMAP applications a selection of examples is presented, e.g., dealing with hygroscopic growth, mixtures of absorbing and non-absorbing particles, the relevance of the size equivalence in case of non-spherical particles, and the variability of volcanic ash microphysics.

The web interface is designed to be intuitive for expert and non-expert users. To support users a large set of default settings is available, e.g., several wavelength-dependent refractive indices, climatologically representative size distributions, and a

parameterization of hygroscopic growth. Calculations are possible for single wavelengths or user-defined sets (e.g., of specific remote sensing application). For expert users more options for the microphysics are available. Plots for immediate visualization of the results are shown. The complete output can be downloaded for further applications. All input parameters and results are stored in the user's personal folder so that calculations can easily be reproduced. The MOPSMAP package is available on request for offline calculations, e.g., when large numbers of different runs for sensitivity studies shall be made.



## 1 Introduction

Aerosol particles in the Earth's atmosphere are important in various ways, for example because of their interaction with electromagnetic radiation and their effect on cloud properties. Consequently aerosol particles are relevant for weather and climate. The temporal and spatial variability of their abundance as well as the variability of their properties is significant which poses

huge challenges in quantifying their effects. This includes the need to establish extended networks of observations using instruments such as photometers (Holben et al., 1998), lidars (Pappalardo et al., 2014), or ceilometers (Wiegner et al., 2014), and the development of models to predict the influence of particles on the state of the atmosphere, see e.g. Baklanov et al. (2014).

Aerosol properties and distributions are often quantified by ground-based and space-borne optical remote sensing and by optical in-situ measurements. These measurements are indirect with respect to microphysical properties (e.g., particle size)

because they measure optical quantities and require the application of inversion techniques to retrieve microphysical properties. Precise knowledge on the link between microphysical and optical properties is needed for the inversion. This link is provided by optical modeling, i.e. the optical properties of particles are calculated based on their microphysical properties. Optical modeling is required also for other applications, e.g., for radiative transfer, numerical weather prediction, and climate modeling. As optical modeling can be very time-consuming it is often inevitable to precalculate optical properties of particles and store

them in a lookup table, which is then accessed by the inversion procedures or subsequent models.

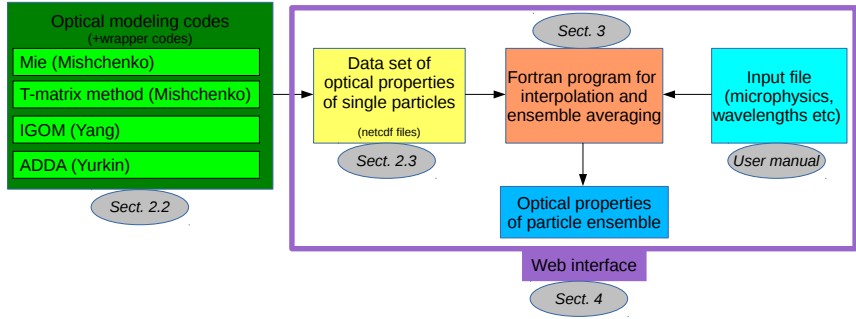

**Figure 1.** Scheme of the MOPSMAP package, including the optical modeling codes applied to create the data set.

In our contribution we describe the MOPSMAP ('Modelled optical properties of ensembles of aerosol particles') package which consists of a data set of pre-calculated optical properties of single aerosol particles, a Fortran program which calculates the properties of user-defined aerosol ensembles from this data set, and a user-friendly web interface for online calculations. Fig. 1 illustrates the overall scheme of the package, including the optical modeling codes (green box) needed once to prepare

the underlying data set. MOPSMAP is either provided as interactive web interface or is provided for offline applications upon request. The former is possible as MOPSMAP is computational very efficient. Compared to other data sets with predefined aerosol components, such as OPAC (Hess et al., 1998), compared to existing online Mie tools such as the one provided by



Prahl (2018), and compared to GUI tools such as MiePlot Laven (2018), MOPSMAP is more flexible with respect to the characteristics of the aerosol ensembles. Moreover, our data set considers not only spherical particles but also spheroids and a small set of irregularly-shaped dust particles. The output includes ASCII tables for further evaluation, netcdf files for direct application in the radiative transfer model uvspec (Emde et al., 2016) and plots for e.g. educational purposes.

5    In Sect. 2, after defining aerosol properties, we describe how existing optical modeling codes were applied (green box in Fig. 1) to create the optical data set of single particles (yellow box). Subsequently, in Sect. 3, we describe the Fortran program (red box) that uses this data set to calculate optical properties of user-defined particle ensembles. The web interface for online application of the MOPSMAP package is introduced in Sect. 4. To demonstrate the potential of MOPSMAP, several applications are discussed in Sect. 5 before we sum up our paper and give an outlook.

## 2    Background and the MOPSMAP data set

The optical properties of a particle with known microphysical properties are calculated by optical modeling. For the creation of the basic data set of MOPSMAP, optical modeling of single particles has been performed. In this section we first define microphysical and optical properties of single particles and then describe how we created the data set using existing optical modeling codes.

15    We emphasize that the data set is, in principle, applicable to the complete electromagnetic spectrum, however we use, for simplicity, the term 'light' and consequently 'optics' instead of more general terms.

### 2.1    Definition of particle properties

The description of particle properties is well-established and can be found in textbooks in detail of variable depth. Thus, we can restrict ourselves to a brief summary of those properties that are of special relevance for MOPSMAP.

20    The microphysical properties of an aerosol particle are described by its shape, size, and chemical composition.

Atmospheric aerosols might be spherical in shape but many types consist of non-spherical particles, often with a large variety of different shapes. Mineral dust (e.g., Kandler et al., 2009) and volcanic ash aerosols (e.g., Schumann et al., 2011b) are important examples for the latter, but for example also pollen, dry sea salt, or soot particles are usually non-spherical. A quite common approach to consider the particle non-sphericity is to approximate the shape with spheroids (Kahn et al., 1997; Dubovik et al., 2006; Wiegner et al., 2009). Spheroids originate from rotation of ellipses about one of their axes. Only one parameter is required to describe the shape of a spheroid. Mishchenko and Travis (1998) use the 'axial ratio' $\epsilon_m$, which is the ratio between the length of the axis perpendicular to the rotational axis and the length of the rotational axis. By contrast, Dubovik et al. (2006) use the 'axis ratio' $\epsilon_d$, defined as the inverse of $\epsilon_m$. Spheroids with $\epsilon_m < 1$, $\epsilon_d > 1$ are called prolate (elongated) whereas spheroids with $\epsilon_m > 1$, $\epsilon_d < 1$ are oblate (flat) spheroids. The aspect ratio $\epsilon'$ is the ratio between the longest



and the shortest axis, i.e. $\epsilon' = \frac{1}{\epsilon_m} = \epsilon_d$ in case of prolate spheroids and $\epsilon' = \epsilon_m = \frac{1}{\epsilon_d}$ in case of oblate spheroids. Spheroids with $\epsilon' = 1$ are spheres.

The size of a particle commonly is described by its radius or its diameter. While this is unambiguous in case of spheres, more detailed specifications are necessary in case of non-spherical particles. Often the size of an equivalent sphere is used for the

description of the size of such particles. The volume-equivalent radius $r_v$ of a particle with volume $V$ (containing the particle mass, i.e. without cavities) is

$$r_v = \sqrt[3]{\frac{3V}{4\pi}}, \tag{1}$$

whereas the cross-section-equivalent radius $r_c$ of a particle with the orientation-averaged geometric cross sectional area $C_{geo}$ is

$$r_c = \sqrt{\frac{C_{geo}}{\pi}}. \tag{2}$$

In case of spheroids, $r_c$ is equal to the radius of a sphere having the same surface area (as used by Mishchenko and Travis (1998)). For the conversion between $r_v$ and $r_c$, the radius conversion factor

$$\xi_{vc} = \frac{r_v}{r_c} = \sqrt[3]{\frac{3\sqrt{\pi}}{4} \frac{V}{C_{geo}^{3/2}}} \tag{3}$$

is used (Gasteiger et al., 2011b). $\xi_{vc}$ is equal to 1 in case of spheres and decreases with increasing deviation from spherical

shape. Another definition of size is given by the radius of a sphere that has the same ratio between volume and geometric cross section as the particle

$$r_{vcr} = \frac{3V}{4C_{geo}} = \xi_{vc}^3 r_c. \tag{4}$$

This definition corresponds to the case 'VSEQU' presented by Otto et al. (2011), to the 'effective radius' in Eq. 5 of Schumann et al. (2011a), and is more sensitive to non-sphericity in terms of $\xi_{vc}$ than $r_v$ or $r_c$. For example, a particle with $r_c = 1$ $\mu$m and

$\xi_{vc} = 0.9$ implies $r_v = 0.9$ $\mu$m and $r_{vcr} = 0.729$ $\mu$m.

For setting up a data set of optical properties for different wavelengths it is highly beneficial to make use of the size parameter

$$x = \frac{2\pi r}{\lambda}. \tag{5}$$

The size parameter $x$ describes the particle size relative to the wavelength $\lambda$. The advantage of using $x$ is that optical properties

($q_{ext}, \omega_0$, and $\mathbf{F}$, as defined below) at a given wavelength are fully determined by its shape, refractive index $m$, and $x$. Equivalent size parameters $x_v$, $x_c$, and $x_{vcr}$ are calculated from the equivalent radii, analogously to Eq. 5.

The chemical composition of a particle determines its complex wavelength-dependent refractive index $m$. The real part $m_r$ determines the speed of light inside the particle and therefore the refraction of waves on the particle surface in the macroscopic


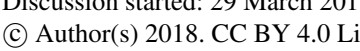


sense. The imaginary part $m_i$ is relevant for the absorption of light inside the particle, whereby an imaginary part of zero corresponds to non-absorbing particles.

The optical properties of a non-spherical particle depend on the orientation of the particle relative to the incident light. In our data set we assume that particles are oriented randomly thus the optical properties are stored as orientation averages

(Mishchenko and Yurkin, 2017).

The orientation-averaged optical properties at a given wavelength are fully described by the extinction cross section $C_{ext}$, the single scattering albedo $\omega_0$, and the scattering matrix $\mathbf{F}(\theta)$ where $\theta$ is the angle by which the incoming light is deflected during the scattering process ('scattering angle'). The extinction cross section $C_{ext}$ can be normalized by the orientation-averaged geometric cross section $C_{geo}$ of the particle giving the extinction efficiency

$$q_{ext} = \frac{C_{ext}}{C_{geo}} = \frac{C_{ext}}{\pi r_c^2} \qquad (6)$$

The single scattering albedo $\omega_0$ is given by

$$\omega_0 = \frac{C_{sca}}{C_{ext}} \qquad (7)$$

where $C_{sca}$ is the scattering cross section.

For the scattering matrix $\mathbf{F}$ of randomly oriented particles we use the notation of Mishchenko and Travis (1998), i.e.

$$\mathbf{F}(\theta) = \begin{bmatrix} a_1(\theta) & b_1(\theta) & 0 & 0 \\ b_1(\theta) & a_2(\theta) & 0 & 0 \\ 0 & 0 & a_3(\theta) & b_2(\theta) \\ 0 & 0 & -b_2(\theta) & a_4(\theta) \end{bmatrix} \qquad (8)$$

with six independent matrix elements. The scattering matrix describes the transformation of the incoming Stokes vector $\mathbf{I^{inc}}$ to the scattered Stokes vector $\mathbf{I^{sca}}$:

$$\mathbf{I^{sca}}(\theta) = \frac{C_{sca}}{4\pi R^2} \mathbf{F}(\theta) \mathbf{I^{inc}} \qquad (9)$$

where the Stokes vectors (van de Hulst, 1981) have the shape

$$\mathbf{I} = \begin{bmatrix} I \\ Q \\ U \\ V \end{bmatrix} \qquad (10)$$

and $R$ is the distance of the observer from the particle. The Stokes vectors $\mathbf{I}$ describe the polarization state of light, with the first element $I$ describing its total intensity. Thus, $\mathbf{F}$ is relevant for the polarization of the scattered light and its first element $a_1$, which is known as the phase function, is important for the angular intensity distribution of the scattered light. The phase





function is normalized such that

$$\int\limits_{0°}^{180°} a_1(\theta) \cdot \sin\theta \cdot d\theta = 2. \tag{11}$$

For many applications it is useful to expand the elements of the scattering matrix using generalized spherical functions (Hovenier and van der Mee, 1983; Mishchenko et al., 2016). The scattering matrix elements at any scattering angle $\theta$ are then

determined by a series of $\theta$-independent expansion coefficients $\alpha_1^l$, $\alpha_2^l$, $\alpha_3^l$, $\alpha_4^l$, $\beta_1^l$, and $\beta_2^l$, with index $l$ from 0 to $l_{max}$, see Eqs. 11-16 in Mishchenko and Travis (1998). $l_{max}$ depends on the required numerical accuracy as well as on the scattering matrix itself. E.g. in case of strong forward scattering peaks (typically occurring at large $x$), $l_{max}$ needs to be larger than in case of more flat phase functions, to get the same accuracy.

The asymmetry parameter $g$ is an integral property of the phase function:

$$g = \frac{1}{2} \int\limits_{0°}^{180°} \cos\theta \cdot a_1(\theta) \cdot \sin\theta \cdot d\theta. \tag{12}$$

$g$ is the average cosine of the scattering angle of the scattered light and is calculated from the expansion coefficients by

$$g = \alpha_1^1/3. \tag{13}$$

## 2.2   Optical modeling of single particles

Depending on the particle type, different approaches are available for calculating particle optical properties. For the creation

of the MOPSMAP optical data set we use in case of spherical particles the well-known Mie theory (Mie, 1908; Horvath, 2009), which is a numerically exact approach over a very broad range of sizes. For spheroids we use the T-matrix method (TMM), which is a numerically exact method but limited with respect to maximum particle size. For larger spheroids not covered by TMM we apply the improved geometric optics method (IGOM). For irregularly-shaped particles the discrete dipole approximation (DDA) is applied.

### 2.2.1   Mie theory

We use the Mie code developed by Mishchenko et al. (2002) for optical modeling of spherical particles. In contrast to the non-spherical particle types described below, we do not store the optical properties of single particles (in a strict sense) because the properties of spheres can be strongly size-dependent which would require a very high size resolution of our data set (e.g., Chýlek, 1990). Instead, we store data averaged over very narrow size bins, allowing us to use a lower size resolution resulting

in a smaller storage footprint of the data set. Actually, for each size parameter grid point $x$, we consider a size parameter bin covering the range from $x/\sqrt{1.01}$ to $x \cdot \sqrt{1.01}$, and apply the Mie code for 1000 logarithmically equidistant sizes within that bin before these results are averaged and stored.



### 2.2.2 T-matrix method (TMM)

We use the extended precision version of the code described by Mishchenko and Travis (1998) for modeling optical properties
of spheroids. To improve the coverage of the particle spectrum ($x$, $\epsilon_m$, and $m$), internal parameter values of the TMM code,
which primarily determine the limits of the convergence procedures, were increased (NPN1 = 290, NPNG1 = 870, NPN4 = 260)

as discussed by Mishchenko and Travis (1998). Though, in general, the TMM provides exact solutions for scattering problems,
non-physical results might be obtained due to numerical problems. To reduce the probability of non-physical results and to
increase the accuracy of the results, the parameter DDELT, i.e. the absolute accuracy of computing the expansion coefficients,
was set to $10^{-6}$ (default $10^{-3}$). In non-converging cases, which happened near the upper limit of the covered size range, the
requirements were relaxed to DDELT = $10^{-3}$. Cases that did not converge even with the relaxed DDELT were not included

in the data set. Nevertheless, some non-physical results were obtained by this approach, for example, $\omega_0 > 1$, or outliers of
otherwise smooth $\omega_0(x)$ or $g(x)$ curves. Thus, for plausibility checks for each particle shape and refractive index, single
scattering albedos $\omega_0$ and asymmetry parameters $g$ were plotted over size parameter $x$ and outliers were recalculated with
slightly modified size parameters. Recalculations with non-physical results were not included in the data set, which reduces
the upper limit of the covered size range for that particular particle shape and refractive index.

### 2.2.3 Improved geometric optics method (IGOM)

Optical properties of large spheroids were calculated with the improved geometric optics method (IGOM) code provided by
Bi et al. (2009); Yang et al. (2007). In general, this approximation is most accurate if the particle and its structures are large
compared to the wavelength. In addition to reflection, refraction, and diffraction by the particle, which are considered by
classical geometric optics codes, IGOM also considers the so-called edge effect contribution to the extinction efficiency $q_{ext}$

(Bi et al., 2009). Classical geometric optics results in $q_{ext}$=2, whereas $q_{ext}$ is variable in case of IGOM. The default settings of
the code were used. The minimum size parameter was selected depending on the maximum size achieved with TMM.

### 2.2.4 Discrete dipole approximation code ADDA

Natural non-spherical aerosol particles, such as desert dust particles, comprise practically an infinite number of particle shapes,
thus it is impossible to cover the full range of shapes in aerosol models. Moreover, the shape of each individual particle is

never known under realistic atmospheric conditions. Consequently, typical irregularities such as flat surfaces, deformations or
aggregation of particles, can be considered only in an approximating way. To enable the user of MOPSMAP to investigate the
effects of such irregularities the properties of six exemplary irregular particle shapes, as introduced by Gasteiger et al. (2011b),
are provided. The geometric shapes were constructed using the object modeling language Hyperfun (Valery et al., 1999). The
first three shapes are prolate spheroids with varying aspect ratios (A: $\epsilon'$=1.4, B: $\epsilon'$=1.8, C: $\epsilon'$=2.4) and surface deformations




according to Gardner (1984). Shape D is an aggregate composed of ten overlapping oblate and prolate spheroids; surface deformations were applied as for shapes A-C. Shape E and F are edged particles with flat surfaces and varying aspect ratio.

The optical properties were calculated with the discrete dipole approximation code ADDA (Yurkin and Hoekstra, 2011). A large number of particle orientations needs to be considered for the determination of orientation-averaged properties. ADDA provides an optional built-in orientation averaging scheme in which the calculations for the required number of orientations is done within a single run. An individual ADDA run using this scheme requires approximately the time for one orientation multiplied with the number of orientations (typically a few hundreds), which can result in computation times of several weeks for large $x$. Because of the long computation times we split them up and performed independent ADDA runs for each orientation. The orientation-averaged properties are calculated in a subsequent step using the ADDA results for the individual orientations.

The computational demand of DDA calculations increases strongly with size parameter $x$, typically with about $x^5$ to $x^6$. Thus, when aiming for large $x$, which is required for mineral dust in the visible wavelength range, it is necessary to find code parameters and an orientation averaging approach that provide a compromise between computation speed and accuracy.

The orientation sampling and averaging is described in Appendix A. The ADDA code allows mainly the following code parameters to be optimized:

– DDA formulation

– Stopping criterion of the iterative solver

– Number of dipoles per wavelength

We estimate the accuracy of the ADDA results by comparing $q_{ext}$, $q_{sca}$, $a_1(0°)$, $a_1(180°)$, and $a_2(180°)/a_1(180°)$ with results obtained using more strict calculation parameters. Accuracy tests are performed for shapes B and C (Fig. 1 of Gasteiger et al. (2011b)), for size parameters $x_v$=10.0, 12.0, 14.4, 17.3, 19.0, and 20.8, and for refractive index $m = 1.52 + 0.0043i$, i.e. 12 single particle cases are considered in total. By comparing the different DDA formulations available in ADDA, it was found that the filtered coupled-dipole technique (ADDA command line parameter "-pol fcd -int fcd"), as introduced by Piller and Martin (1998) and applied by Yurkin et al. (2010), offers the best compromise between computation speed and accuracy of modeled optical properties. Using a stopping criterion for the iterative solver of $10^{-4}$ instead of $10^{-3}$, has only negligible influence on orientation-averaged optical properties ($< 0.1\,\%$) but requires approximately $30\,\%$ more computation time; thus, we used $10^{-3}$ for the ADDA calculations to create our data set. The extinction efficiency $q_{ext}$ and the scattering efficiency $q_{sca}$ change in all cases by less than $0.3\,\%$ if a grid density of 16 dipoles per wavelength is used instead of 11. The maximum relative changes due to the change in dipole density are 0.2% for $a_1(0°)$, 1.7% for $a_1(180°)$, and 1.9% for $a_2(180°)/a_1(180°)$. Because of the large difference in computation time, which is about a factor 3-4, and the low loss in accuracy, 11 dipoles per wavelength were selected for the MOPSMAP data set.

To test the accuracy of the selected orientation averaging scheme, optical properties were calculated using a denser grid of orientation grid points ($\beta_e$ and $\gamma_e$, as defined in Appendix A, but with a step size of 5°); these calculations consider about



**Table 1.** Microphysics of spheres and spheroids considered in the MOPSMAP data set. IGOM was not applied to $m \leq 1.0$.

| method | Mie | TMM | IGOM |
|---|---|---|---|
| particle shape | spheres | oblate and prolate spheroids | |
| | | $\epsilon'$=1.2, 1.4, ..., 3.0, 3.4, 3.8, ..., 5.0 | |
| size parameter | $10^{-6} < x_c < 1005$ | $10^{-6} < x_c < (5-125)$ | $(5-125) < x_c < 1005$ |
| | $\frac{x_{i+1}}{x_i} = 1.01$ | $\frac{x_{i+1}}{x_i} = 1.05$ | $\frac{x_{i+1}}{x_i} = 1.10$ |
| | size bins | single size | single size |
| $m_r$ | 0.1, 0.2, ..., 0.9, 1.0, 1.04, 1.08, ..., 1.68, 1.76, ..., 2.0, 2.2, ..., 3.0 | | |
| $m_i$ | 0, 0.0005375, 0.001075, 0.0015203, 0.00215, | | |
| | 0.0030406, 0.0043, 0.0060811, 0.0086, 0.0121622, | | |
| | 0.0172, 0.0243245, 0.0344, 0.0486490, 0.0688, | | |
| | 0.0972979, 0.1376, 0.2752, 0.5504, 1.1008, 2.2016 | | |

**Table 2.** Microphysics of irregular shapes considered in the MOPSMAP data set.

| | |
|---|---|
| particle shape | shapes A-F, Fig. 1 of Gasteiger et al. (2011b) |
| size parameter | $10^{-3} < x_v < 27.5; \frac{x_{i+1}}{x_i} \approx 1.10;$ single size |
| $m_r$ | 1.48, 1.52, 1.56, 1.60 |
| $m_i$ | 0, 0.00215, 0.0043, 0.0086, 0.0172, 0.0344, 0.0688 |

12 times more orientations than the calculations for our data set. Maximum deviations of 0.6 % for $q_{ext}$ and $q_{sca}$ are found. $a_1(0°)$, $a_1(180°)$, and $a_2(180°)/a_1(180°)$ change by not more than 0.9%, 6.5%, and 12%, respectively.

In summary, ADDA with the filtered coupled-dipole technique, 11 dipoles per wavelength, and a stopping criterion for the iterative solver of $10^{-3}$ was used for optical modeling of the irregularly-shaped particles in our data set together with the orientation averaging scheme described in Appendix A. Tests demonstrate that the modeling accuracy is mainly determined by the applied orientation averaging scheme.

## 2.3 Optical data set

Using the codes with the settings described above, a data set of modeled optical properties of single particles in random orientation was created. The complete data set requires about 42 gigabytes of storage capacity. For spheres we stored, instead of single particle properties, averages over narrow size bins as described above. An overview over the wide range of sizes, shapes, and refractive indices of the particles in the data set is given in Tables 1 and 2. For each combination of refractive index





and shape a separate netcdf file was created, e.g., 'spheroid_0.500_1.5200_0.008600.nc' for spheroids with $\epsilon_m = 0.5$ (prolate with $\epsilon' = 2.0$) and $m = 1.52 + 0.0086i$. Each file contains the optical properties on a grid of size parameters.

For spheres and spheroids the minimum size parameter is set to $10^{-6}$, and the maximum size parameter is set to $x \approx 1005$ to cover $r_c = 40$ $\mu$m at $\lambda = 250$ nm. The size increment is 1% (i.e. $x_{i+1}/x_i = 1.01$) in case of spheres, 5% in case of TMM
spheroids, and 10% for IGOM spheroids. In case of spheroids, the TMM is applied up to the largest possible size parameter with the approach described in Sect. 2.2.2. The maximum size parameter for TMM determines the lowest size parameter for which IGOM is applied. The first IGOM size parameter is between 0 and 10% larger than the maximum TMM size parameter. The transition size parameter between TMM and IGOM is in the range $5 < x < 125$, strongly depending on $m$ and particle shape. The TMM and IGOM results for spheroids are merged into a single netcdf file covering the complete size range from
$x = 10^{-6}$ to $x \approx 1005$, which is sufficient for most applications. For example, for prolate spheroids with $\epsilon' = 1.8$ and $m = 1.56 + 0i$, the size range from $x = 10^{-6}$ to $x = 88.22$ is covered by TMM, while IGOM starts at $x = 89.54$. Since IGOM is an approximation unrealistic jumps of optical properties may occur at the transition size parameter. For typical mineral dust ensembles in the visible spectrum, particles in the TMM range contribute more than 90% to the total extinction. IGOM was not applied to $m_r < 1.04$, thus the size parameter range is limited to the TMM range for these refractive indices. A step of 0.04
was selected for the $m_r$ grid in the most relevant range and a wider $m_r$ step for the less common ranges. Development of the data set started with $m_i = 0.0043$ and beginning from this value, $m_i$ was increased and decreased in steps of factor $\sqrt{2}$. Below $m_i = 0.001$ and above $m_i = 0.1$ the step width is a factor of 2.

The optical data for the irregularly-shaped particles (Tab. 2) is limited to $x_v \leq 27.5$ because of the huge computation requirements for optical modeling of large particles. Nonetheless, this range already covers the most important range for many
applications. E.g., at $\lambda = 1064$ nm particles up to $r_v = 4.6$ $\mu$m are covered. The $m$ grid for the irregularly-shaped particles is limited to the most relevant range for desert dust in the visible spectrum and the $m_i$ step is set to a factor of 2. The quantification of the conversion factor $\xi_{vc}$ of the six irregular shapes requires the determination of their orientation averaged geometric cross sections which is done numerically.

The optical properties stored for each particle are the extinction efficiency $q_{ext}$, the scattering efficiency $q_{sca}$, and the ex-
pansion coefficients $\alpha_1^l$, $\alpha_2^l$, $\alpha_3^l$, $\alpha_4^l$, $\beta_1^l$, and $\beta_2^l$ of the scattering matrix. The ADDA and the IGOM code provide the angular-resolved scattering matrix elements which we converted to the expansion coefficients stored in the data set following the method described by Hovenier and van der Mee (1983); Mishchenko et al. (2016). We optimized the expansion coefficients for accurate scattering matrices at 180°, which probably is the most error sensitive angle. As a by-product lidar applications will certainly benefit from this optimization.

In case of asymmetric shapes in random orientation, the scattering matrix has 10 independent elements as discussed by van de Hulst (1981). By using only six elements of $\mathbf{F}$ (Eq. 8) in our data set we implicitly assume that each irregular model particle (shapes A-F) occurs as often as its mirror particle, which is formed by mirroring at a plane (van de Hulst, 1981).





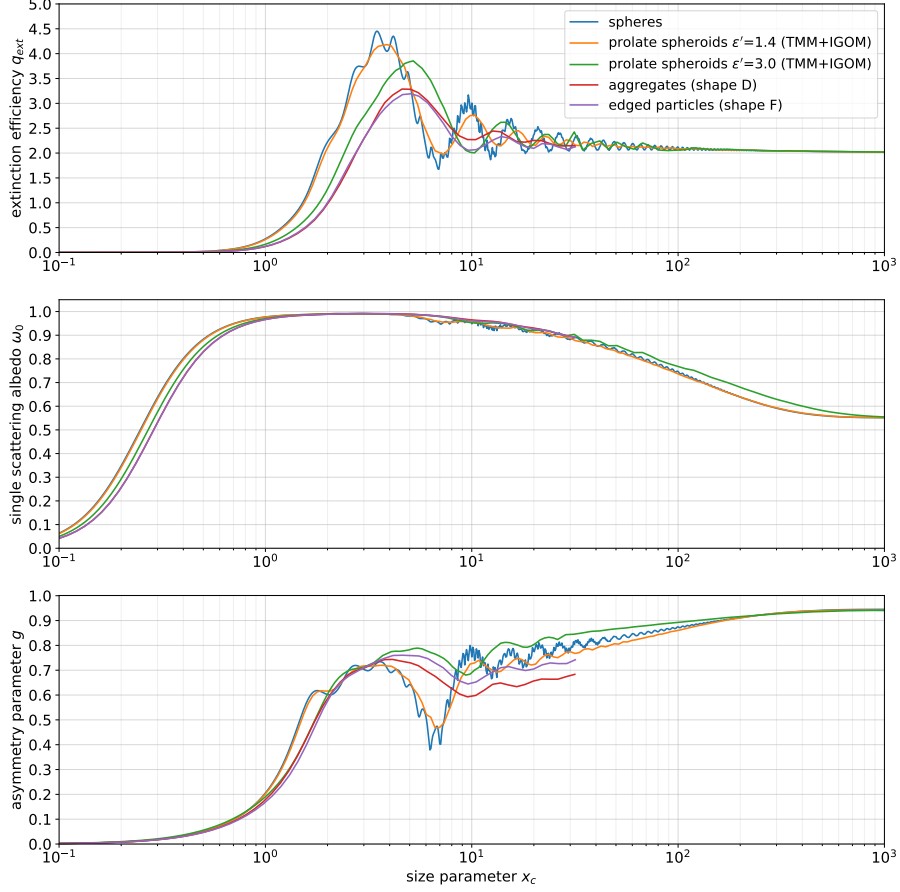

**Figure 2.** Optical properties of single particles (or narrow size bins in case of spheres) with fixed refractive index $m = 1.56 + 0.00215i$ as function of size parameter. The different colors denote different particle shapes. The upper panel shows the extinction efficiency $q_{ext}$, the middle panel the single scattering albedo $\omega_0$, and the lower panel the asymmetry parameter $g$.

Fig. 2 shows an example of the MOPSMAP optical data set. The refractive index is set to $m = 1.56 + 0.00215i$, which is representative for desert dust particles at visible wavelengths. The properties of spherical particles are shown in blue, while the properties of prolate spheroids with $\epsilon' = 1.4$ and $3.0$ are shown in orange and green, respectively. Red and violet lines denote irregularly-shaped particles D and F, respectively. The upper panel shows the extinction efficiency $q_{ext}$ as function of cross-
5  section-equivalent size parameter $x_c$. The general shape of the $q_{ext}(x_c)$-curve is similar for the different shapes; nonetheless, with increasing deviation from spherical shape, the amplitudes of the oscillations of $q_{ext}(x_c)$ get smaller and a shift of the maximum $q_{ext}$ towards larger $x_c$ is found. The middle panel shows the single scattering albedo $\omega_0$ for the same particles as the upper panel. For particle sizes comparable to the wavelength, $\omega_0$ reaches maxima with values of about 0.991, almost independent of particle shape. $\omega_0$ approaches a value of about 0.551 at $x_c \approx 1000$ for spheres and spheroids. The lower panel
10  shows the asymmetry parameter $g$. When the particle size becomes comparable to the wavelength, $g$ increases and oscillates





as function of $x$ with the strongest oscillations occurring in case of spheres. There is some shape dependence of $g$ for $x > 5$, in particular the aggregate shape results in systematically smaller $g$ than the other shapes for $x > 10$. The transition from the numerically exact TMM to the IGOM approximation occurs at $x \approx 125$ for $\epsilon' = 1.4$ (orange line) and at $x \approx 27$ for $\epsilon' = 3.0$ (green line) and is quite smooth.

## 3  MOPSMAP Fortran program

In this section the basic characteristics of the MOPSMAP Fortran program to calculate optical properties of particle ensembles is described. Besides a modern Fortran compiler, e.g., gfortran 6 or above, the netCDF Fortran development source code is required to build the executable. The computation time and memory requirements depend on the ensemble complexity and the number of wavelengths but in general are low for state-of-the-art personal computers. The Fortran code and the data set are

available upon request. A web interface (see Sect. 4) provides online access to most of the functionality of the Fortran program without the requirement to download the code and the data set.

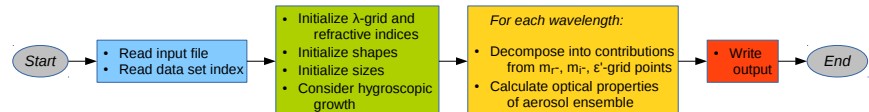

**Figure 3.** Simplified flow chart of the MOPSMAP Fortran program.

Within each MOPSMAP run the optical properties of a specific user-defined ensemble are calculated at a user-defined wavelength grid. The ensemble microphysics and the wavelength grid are defined in an input file. The details about the options available for the input file are described in a user manual which is provided together with the code.

Fig. 3 shows a flow chart of the MOPSMAP Fortran program. The program is initialized by reading an input file and a data set index. The latter contains information on the refractive index and shape grid and the size parameter ranges covered by the data set. Then, all information required for the optical modeling is initialized, for example the set of wavelengths, the refractive indices as function of wavelength, shape distributions, and the effect of the hygroscopic growth, before the optical calculations are performed for each wavelength, as described in the following.

### 3.1  Calculation of optical properties of particle ensembles

Usually aerosol particles occur as ensembles of particles of different size, refractive index, and/or shape. The different particles contribute to the optical properties of the ensemble. Assuming that the distance between the particles is large enough so that interaction of light with each particle occurs without influence of any other particle ('independent scattering'; van de Hulst, 1981), the contribution of each particle can be added as described below.





In MOPSMAP particle ensembles are composed of one or more independent modes (the terms 'mode' and 'component' are often used synonymously in the literature). Each mode in MOPSMAP is characterized by particle size, shape, and refractive index, whereby each property can be described as a fixed value or as a distribution (see below). As these parameters do not necessarily correspond to the grid points of the MOPSMAP data set, interpolation and decomposition into different contribu-

tions is performed. In case of fixed values of $m_r$, $m_i$, and $\epsilon'$ for a mode, the mode is decomposed into eight contributions. This is done as follows: If the next grid points of the data set that are smaller and larger are $m_{r,i}$ and $m_{r,i+1}$ respective for the real part of the refractive index, $m_{i,j}$ and $m_{i,j+1}$ for the imaginary part, and $\epsilon'_k$ and $\epsilon'_{k+1}$ for the aspect ratio with

$$m_{r,i} \leq m_r < m_{r,i+1}, \quad m_{i,j} \leq m_i < m_{i,j+1} \quad \text{and} \quad \epsilon'_k \leq \epsilon' < \epsilon'_{k+1}, \tag{14}$$

the weight $w$ of each grid point is related to the normalized distances from the grid points, e.g.,

$$w_{mr,i} = \frac{m_{r,i+1} - m_r}{m_{r,i+1} - m_{r,i}} \quad \text{and} \quad w_{mr,i+1} = \frac{m_r - m_{r,i}}{m_{r,i+1} - m_{r,i}} \tag{15}$$

or analogously for the weights of the $m_i$ and $\epsilon'$ grid points. Finally the weights for each of the eight contributions are calculated as the products of the three corresponding weights; for example, the contribution from the $m_{r,i+1}$, $m_{i,j}$, and $\epsilon'_{k+1}$ grid point is weighted with $w = w_{mr,i+1} \cdot w_{mi,j} \cdot w_{\epsilon',k+1}$. This approach basically results in a linear interpolation of extensive properties between the $m_r$, $m_i$, and $\epsilon'$ grid points of the data set. The error of the interpolation of the user-specified values between the

grid points of the data set is discussed in Sect. 3.3. In case a $\epsilon'$ distribution is given for a mode, contributions from the required range of $\epsilon'$ grid points are considered and weighted according to the $\epsilon'$ distribution. The decomposition into contributions is done independently for each mode.

Because of the limited size range of irregularly-shaped particles (see above) they require a special treatment: For cases when the size range of a user-defined mode exceeds size parameter $x_v$=27.5, which is the upper limit for these particles in the

data set, a MOPSMAP option is available which substitutes irregularly-shaped particles above this $x_v$-limit with other particle shapes, spherical or non-spherical, as selected by the user. As a consequence, the particle shape of that mode becomes size- and wavelength-dependent.

The optical properties of the particle ensemble are calculated for each wavelength by summation over extensive properties of all particles. Assume that the user-defined ensemble is decomposed into $J$ contributions as just described. Each contribution

has a size distribution $n_j(r)$, i.e. a particle number concentration per particle radius interval from $r$ to $r+dr$, in the range from $r_{min,j}$ to $r_{max,j}$, which is obtained by multiplying the user-defined size distribution of the mode with the weights obtained during the decomposition. The extinction coefficient $\alpha_{ext}$ and the scattering coefficient $\alpha_{sca}$ are calculated by

$$\alpha_{ext} = \sum_{j=1}^{J} \left( \int_{r_{min,j}}^{r_{max,j}} C_{ext,j}(r) \cdot n_j(r) \cdot dr \right) \quad \text{and} \quad \alpha_{sca} = \sum_{j=1}^{J} \left( \int_{r_{min,j}}^{r_{max,j}} C_{sca,j}(r) \cdot n_j(r) \cdot dr \right). \tag{16}$$





The expansion coefficients need to be weighted with $C_{sca,j}(r)$, for example $\alpha_1^l$ of a particle ensemble is calculated by

$$\alpha_1^l = \frac{1}{\alpha_{sca}} \cdot \sum_{j=1}^{J} \left( \int_{r_{min,j}}^{r_{max,j}} \alpha_{1,j}^l(r) \cdot C_{sca,j}(r) \cdot n_j(r) \cdot dr \right). \tag{17}$$

For the integration of extensive properties over the size distribution, we apply the trapezoidal rule, which assumes linearity between the $r$ grid points.

The size distribution $n(r) = \frac{dN}{dr}$ for each mode can be specified in various ways. The MOPSMAP user can either specify a single size, apply size distribution tables in ASCII format, or apply a size distribution parameterization. The following size distribution parameterizations are available:

1.  $n(r) = \frac{1}{\sqrt{2\pi}} \frac{n_0}{\ln\sigma} \frac{1}{r} \exp\left[ -\frac{1}{2} \left( \frac{\ln r - \ln r_{mod}}{\ln\sigma} \right)^2 \right]$  —  log-normal distribution

2.  $n(r) = ar^\alpha \exp\left( -br^\gamma \right)$  —  modified gamma distribution, (Deirmendjian, 1964)

3.  $n(r) = a\exp\left( -br \right)$  —  exponential distribution, $\alpha=0$, $\gamma = 1$

4.  $n(r) = ar^\alpha$  —  power law distribution, Junge distribution, $b = 0$, (Deirmendjian, 1964)

5.  $n(r) = ar^\alpha \exp\left( -br \right)$  —  gamma distribution, $\gamma = 1$, (Twomey, 1977)

with $r_{mod}$ the mode radius, $\sigma$ a dimensionless parameter for the relative width of the distribution , and $n_0$ the total number density (in the range from $r_{min} = 0$ to $r_{max} = \infty$) of the log-normal distribution. For the subsequent size distributions,

parameters $a$, $\alpha$, $b$, and $\gamma$ are positive and $a$ controls the scaling of total number density while $\alpha$, $b$, and $\gamma$ are relevant for the shape of the size distributions. The exponential distribution, power law distribution, and the gamma distribution, are a subset of the modified gamma distribution with the specific parameter values as given above (see also Petty and Huang, 2011).

The particle shape can be specified independently for each mode and is, within each mode, independent of size and refractive index. In case of spheroids, either a fixed aspect ratio $\epsilon'$ or an aspect ratio distribution is used. The latter can be given as a table

in an ASCII file or it can be parameterized by a modified log-normal distribution (Kandler et al., 2007)

$$n(\epsilon') = \frac{dN}{n_0 \cdot d\epsilon'} = \frac{1}{\sqrt{2\pi}\sigma_{ar}(\epsilon'-1)} \exp\left[ -\frac{1}{2} \left( \frac{\ln(\epsilon'-1) - \ln(\epsilon_0'-1)}{\sigma_{ar}} \right)^2 \right] \tag{18}$$

with parameters $\epsilon_0'$ for the location of the maximum of $n(\epsilon')$ and $\sigma_{ar}$ for the width of the distribution.

The refractive index of each mode can either be wavelength-independent or specified as function of wavelength in an ASCII file. In addition, it is possible to specify for each mode a non-absorbing fraction $\mathcal{X}$. If $\mathcal{X} > 0$, the mode is divided, for all

sizes and shapes, into a non-absorbing ($m_{i,1}=0$, relative abundance $\mathcal{X}$) and an absorbing fraction ($m_{i,2}=m_i/(1-\mathcal{X})$, relative abundance $1-\mathcal{X}$). As a consequence, the average $m_i$ over all particles of the mode remains equal to the $m_i$ as specified by the

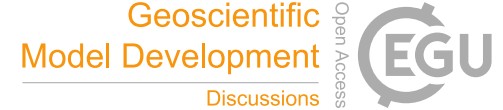


user. This non-absorbing fraction approach can be used as a parameterization of the refractive index variability within desert dust ensembles as described by Gasteiger et al. (2011b) and below in Sect. 5.6.

For the hygroscopic particle growth the following parameterization (Petters and Kreidenweis, 2007; Zieger et al., 2013)

$$\frac{r_{wet}(\text{RH})}{r_{dry}} = \left(1 + \kappa \cdot \frac{\text{RH}}{1 - \text{RH}}\right)^{\frac{1}{3}} \tag{19}$$

is implemented in MOPSMAP where RH is the relative humidity and $\kappa$ the hygroscopic growth parameter of the particles of each mode. This equation describes the ratio between the size of the particle at a given RH and the size of the particle in a dry environment (RH=0%). The parameterization implies that this ratio is independent of size, thus for example in case of a log-normal size distribution, $r_{min}$, $r_{max}$, and $r_{mod}$ are multiplied with this ratio, while the relative width $\sigma$ of the distribution is not modified. This is the usual approach though modal representations of aerosol size distributions may also predict higher

moments (Binkowski and Shankar, 1995; Zhang et al., 2002), and thus $\sigma$ can be a prognostic variable as well. The refractive index is modified by the taken up water following the volume weighting rule. Both RH and $\kappa$ can be chosen by the user. This parameterization is valid for particles with $r > 40nm$ where the Kelvin effect can be neglected (Zieger et al., 2013). It is worth noting that this parameterization differs from the relative humidity dependence implemented in OPAC which was adapted from Hänel and Zankl (1979).

## 3.2 Output of Fortran program

As output of MOPSMAP the following properties of aerosol ensemble are available. Redundant properties, such as the lidar-related properties, are available to facilitate the usage of the results.

- Extinction coefficient $\alpha_{ext}$ $[m^{-1}]$

- Single scattering albedo $\omega_0$

- Asymmetry parameter $g$

- Effective radius $r_{eff} = \frac{\int r^3 n(r) dr}{\int r^2 n(r) dr}$ $[\mu m]$

- Number density $n$ $[m^{-3}]$ (number of particles per atmospheric volume)

- Cross section density $a$ $[m^{-1}]$ (particle cross section per atmospheric volume)

- Volume density $v$ (particle volume per atmospheric volume)

- Mass concentration $M$ $[gm^{-3}]$ (particle mass per atmospheric volume)

- Expansion coefficients ($\alpha_1$ to $\beta_2$) for elements of scattering matrix

- Scattering matrix elements ($a_1$ to $b_2$) at user defined angle grid

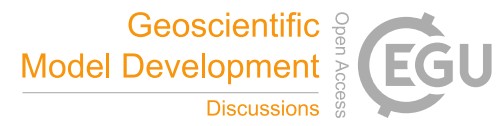

- – Volume scattering function $\widetilde{a}_1 = \frac{\alpha_{ext} \cdot \omega_0}{4\pi} \cdot a_1 \ [m^{-1} sr^{-1}]$ at user defined angle grid

- – Backscatter coefficient $\beta = \frac{\alpha_{ext} \cdot \omega_0}{4\pi} \cdot a_1(180°) \ [m^{-1} sr^{-1}]$

- – Lidar ratio $S = \frac{4\pi}{\omega_0 a_1(180°)} \ [sr]$

- – Linear depolarization ratio $\delta_l = \frac{a_1(180°) - a_2(180°)}{a_1(180°) + a_2(180°)}$

5 - – Ångström exponents $\text{AE}_\zeta = -\frac{\log \frac{\zeta(\lambda_1)}{\zeta(\lambda_2)}}{\log \frac{\lambda_1}{\lambda_2}}$ for $\zeta \in \{\alpha_{ext}, \alpha_{sca}, \alpha_{abs}, \beta\}$

- – Extinction to mass conversion factor $\eta = \frac{M}{\alpha_{ext}} \ [gm^{-2}]$

- – Mass to backscatter conversion factor $Z = \frac{\beta}{M} \ [m^2 sr^{-1} g^{-1}]$

Scattering matrix elements and the quantities derived from them are calculated from the expansion coefficients. Wavelength-independent properties $r_{eff}$, $n$, $a$, $v$, and $M$, are calculated for each wavelength to demonstrate the numerical accuracy of the 10 integration.

The results are available in ASCII and in netcdf format. The format of the program output is described in the user manual. The netcdf output files can be read by the radiative transfer model uvspec, which is included in libRadtran (Mayer and Kylling, 2005; Emde et al., 2016).

### 3.3 Interpolation and sampling error

15 Due to the limited size resolution in the data set and required interpolations between refractive index and aspect ratio grid points, deviations from exact model calculations for specific microphysical properties occur. As examples, Fig. 4 illustrates deviations introduced for single particle properties, while Tab. 3 shows deviations for particle ensembles.

On the left-hand side of Fig. 4 effects of the limited size resolution on the extinction efficiency $q_{ext}$ and the asymmetry parameter $g$ are shown for non-absorbing spheres and spheroids with $m$=1.52. In particular for spheres with $x > 10$ deviations 20 for single particles can be considerable because of small-scale features that are not resolved in the data set. In case of spheres these features are implicitly considered in the data set by storing the average over 1000 sizes within each size bin as described above. In case of spheroids the data set contains properties calculated for single sizes which may not be fully representative for close-by sizes. However, since the small-scale features are much weaker for spheroids than for spheres, the average deviation for spheroids is much smaller than for spheres.

25 On the right-hand side of Fig. 4 effects due to the required interpolation between the refractive index grid points are illustrated for spheres with $m$=1.54+0.005i. While the red lines show the properties calculated from the data set, the black lines show Mie calculations done explicitly for $m$=1.54+0.005i with the same size grid as used in the data set. The comparison illustrates that MOPSMAP calculates optical properties on average correctly but some smaller scale features are lost: for example, the





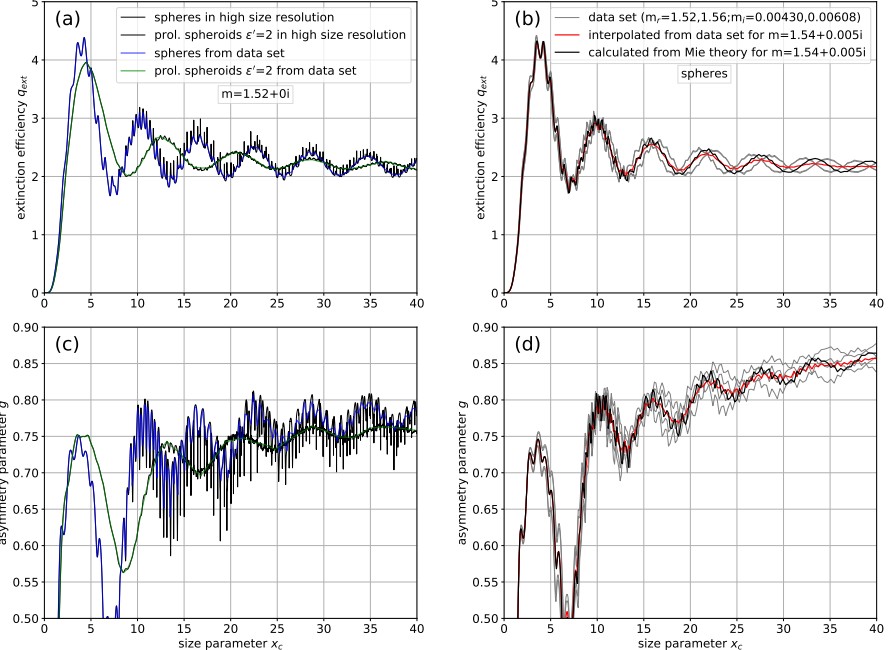

**Figure 4.** Examples illustrating the effect of the limited size resolution of the MOPSMAP data set (left-hand side) and the effect of the interpolation between the refractive index grid points of the data set (right-hand side); extinction efficiencies $q_{ext}$ (upper panels) and asymmetry parameters $g$ (lower panels) as functions of the size parameter from $x$=0 to $x$=40 are compared; in a) and c) the high size resolution calculations (grey lines) were performed with linear $x$ steps of 0.002 in case of spheres and 0.01 in case of spheroids; in b) and d) the red lines show properties calculated for $m$=1.54+0.005i with MOPSMAP by interpolation between refractive indices included in the data set (their data shown as four thin grey lines), and the black lines show for comparison the properties calculated by Mie theory explicitly for $m$=1.54+0.005i (using the same $x$ grid as used by the data set).

extinction efficiency $q_{ext}(x)$ in the size parameter range from 20 to 40 is dampened compared to the Mie calculation for $m$=1.54+0.005i because of the interference of the $q_{ext}(x)$ curves for $m_r$=1.52 and $m_r$=1.56 (see grey lines in Fig. 4b; note that curves for different $m_i$ lie almost on top of each other).

For other size ranges, refractive indices, and optical quantities, the effects on the single particle properties are in principle similar but they may vary in magnitude.

Tab. 3 investigates the sampling and interpolation errors for a mono-modal log-normal size distribution with a typical width of $\sigma$=2.0. The effective radius is $r_{eff}$=1.44$\mu$m which a typical value for transported desert aerosol. Sizes up to $r_{max}$=4$\mu$m, which corresponds to size parameter $x_c = 40$ at $\lambda$=0.62832$\mu$m, are considered. The left half of Tab. 3 compares optical properties calculated from the MOPSMAP data set (columns 'data set') with properties calculated using a high size resolution (columns 'highres'), the same resolutions as displayed in Fig. 4a. For spheres the results are equal up to at least the fourth digit.



**Table 3.** Optical properties calculated for a log-normal mode with $r_{mod}$=0.5$\mu$m, $\sigma$=2.0, $r_{min}$=0.001$\mu$m, and $r_{max}$=4$\mu$m at $\lambda$=0.62832$\mu$m. Two cases of particle shapes are considered: spheres and prolate spheroids with $\epsilon'$=2.0. Columns 'data set' contain values calculated using MOPSMAP with the data set described in Sect. 2.3. For comparison, the same properties are calculated in columns 'highres' using high size resolution and in columns 'explicit' using Mie theory or TMM explicitly at $m$=1.54+0.005i.

| | size sampling example for $m$=1.52+0i | | | | $m$-interpolation example for $m$=1.54+0.005i | | | |
|---|---|---|---|---|---|---|---|---|
| | spheres | | spheroids | | spheres | | spheroids | |
| | data set | highres | data set | highres | data set | explicit | data set | explicit |
| $\alpha_{ext}$ [km$^{-1}$] | 4.808 | 4.808 | 4.863 | 4.861 | 4.793 | 4.793 | 4.844 | 4.846 |
| $\omega_0$ | 1.0000 | 1.0000 | 1.0000 | 1.0000 | 0.8845 | 0.8840 | 0.8892 | 0.8886 |
| $g$ | 0.7045 | 0.7045 | 0.7018 | 0.7021 | 0.7331 | 0.7332 | 0.7382 | 0.7380 |
| $S$ [sr] | 10.52 | 10.52 | 42.75 | 42.30 | 13.13 | 13.36 | 58.25 | 58.78 |
| $\delta_l$ | 0.0000 | 0.0000 | 0.3063 | 0.2986 | 0.0000 | 0.0000 | 0.2502 | 0.2502 |

In case of prolate spheroids with $\epsilon'$=2.0, deviations are found for the forth digit of $\alpha_{ext}$ and $g$. For the lidar-related quantities $S$ and $\delta_l$ the differences are larger with the relative deviation of $\delta_l$ being 2.6%. These differences are caused by the high sensitivity of lidar-related quantities and it is expected that deviations become smaller when shape distributions or wider size distributions are applied.

5  The right half of Tab. 3 demonstrates the effect of the $m$-interpolation for an exemplary $m$=1.54+0.005i. MOPSMAP calculations (columns 'data set') are compared to results obtained using explicitly this refractive index in the Mie and TMM calculations. While the effect of the $m$-interpolation is very small for $\alpha_{ext}$, $g$, and $\delta_l$, it is slightly larger for $\omega_0$ and $S$. The maximum relative effect is found for the lidar ratio $S$ of spheres with a deviation of 1.7%.

These comparisons demonstrate that deviations found for single particles are largely smoothed out in case of particle ensem-
10 bles due to the averaging over a large number of single particles.

## 4 MOPSMAP web interface

A web-interface is provided as part of MOPSMAP at https://mopsmap.net. It was designed to be intuitive for expert and non-expert users, e.g., for the demonstration of sensitivities of optical properties on microphysical properties in the framework of lectures, but also for a lot of scientific problems as outlined in the following section. The web interface is written in PHP and
15 uses the SQLite library. After the registration as a user, online calculations of optical properties of a large range of particle ensembles can be performed. Input and output can be defined by the user; for non-expert users a lot of default ensembles representative for specific climatological conditions are already available. The input parameters primarily include the micro-



physical properties of the particles. The particles' microphysics are described by up to four components (each described by an individual log-normal size distribution), the wavelength dependence of the refractive index and the shape. Any log-normal size distribution can be used; to facilitate the usage (e.g., for non-expert users) the aerosol components from the OPAC data set (Hess et al., 1998), e.g., "mineral coarse mode", "water-soluble", or "soot", are already included. The same is true for the

ten "aerosol types" defined in OPAC, e.g., "continental clean", "urban" or "maritime polluted", consisting of a combination of components. Calculations can be made for a single wavelength, for wavelength ranges or a pre-scribed wavelengths-set (e.g., for a typical aerosol lidar or a AERONET sun photometer). Moreover, the user can define own wavelengths-sets, e.g., for a specific radiometer. The relative humidity is selected by the user and it is effective for all hygroscopic components according to Eq. 19. The hygroscopic growth of the OPAC components in MOPSMAP differs from the original OPAC version (Hess

et al., 1998); it follows the $\kappa$-parameterization with the values proposed by Zieger et al. (2013). In the 'expert user mode' the flexibility is further increased: the number of components can be larger than four, and the size distribution can be given as discrete values on a user-defined size grid.

The output comprises the complete set of optical properties as described in Section 3.2. It can be downloaded for further applications and includes ASCII tables as well as a netcdf-file as required for radiative transfer calculations using uvspec of the

widely used Libradtran package (Emde et al., 2016). To provide an immediate overview over the results, the most important parameters, such as extinction coefficient ($\alpha_{ext}$), single scattering albedo ($\omega_0$), asymmetry parameter ($g$), Ångström exponent (AE), or lidar ratio ($S$), are displayed as tables when the calculations have been completed. In addition plots of the results as function of wavelength and scattering angle are shown as selected by the user.

All results are stored in the user's personal folder so that all calculations can be reproduced. Furthermore, all calculations

can also easily be rerun with a slightly modified input parameter set.

## 5  Applications

In this section a selection of examples is presented to demonstrate the wide range of applications of MOPSMAP. Many of them can be performed by using the web interface. Some examples need a local version of MOPSMAP alongside with scripts that repeatedly call the Fortran program. These scripts are written in Python and can be requested together with the MOPSMAP

package.

It is worth mentioning that numerous studies demonstrate the need for optical modeling of aerosol ensembles, thus illustrating the range of possible applications of MOPSMAP. Moreover, optical modeling is essential for many different related modeling activities. It is required, for example, for closure experiments (consistency checks between different measurement methods involving an aerosol model, e.g., Wiegner et al., 2009; Gasteiger et al., 2011b; Müller et al., 2012; Bell et al., 2013;

Ma et al., 2014; Zieger et al., 2014; Düsing et al., 2018), radiative transfer studies (e.g., Otto et al., 2009; Emde et al., 2010), inversion of remote sensing measurements (e.g., Dubovik et al., 2006; Gasteiger et al., 2011a; Müller et al., 2016), inversion





of in-situ data (e.g., Weinzierl et al., 2009; Szymanski et al., 2009; Kassianov et al., 2014), aerosol layer visibility simulations (e.g., Weinzierl et al., 2012), dynamic aerosol transport models (e.g., Heinold et al., 2007; Balzarini et al., 2015), aerosol characterization (e.g., Gasteiger et al., 2017; Che et al., 2018; Zhuang et al., 2018), and solar energy (e.g., Polo et al., 2016; Kosmopoulos et al., 2017).

## 5.1 Effect of hygroscopicity

The first example of applications deals with hygroscopic growth. If aerosol particles are hygroscopic their microphysical and optical properties change with relative humidity RH. Fig. 5 shows how optical properties of the 10 OPAC aerosol types (Hess et al., 1998), which contain up to four components, some of which being hygroscopic, change with RH. These calculations were performed using the MOPSMAP web interface, where the OPAC aerosol types are available as pre-defined ensembles and the relative humidity can be chosen by the user. MOPSMAP considers the hygroscopic effect by application of the $\kappa$-parameterization (Eq. 19) which differs from the RH dependency implemented in OPAC.

The upper row of Fig. 5 shows the normalized extinction coefficient of the different types (indicated by color) at three wavelengths $\lambda$ (each in a subplot) calculated for RH values of 0%, 50%, 70%, 80%, and 90%. The extinction at all $\lambda$ is normalized to the extinction at RH=0% and $\lambda = 532$ nm. As a consequence, the differences between the columns illustrate the wavelength dependency of the extinction, while changes with RH illustrate the hygroscopic effects. For example, for the desert aerosol type (orange color), the wavelength dependency is low, which is related to the large size of the dominant mineral particles, and the hygroscopic effect is relatively weak because mineral particles are hygrophobic. By contrast, for maritime (bluish colors) and antarctic types (purple color), the wavelength dependence is stronger and the hygroscopic effect is strong because of the domination by highly hygroscopic sulfate and sea salt particles. For the continental as well as the urban and arctic types, the wavelength dependence is even stronger while the hygroscopic effect is weaker, which may be explained by strong contributions from the soot and water-soluble components which contain quite small particles with $\kappa$ values significantly smaller than the $\kappa$ values of sea salt particles.

The single scattering albedo $\omega_0$ is shown in the second row of Fig. 5. $\omega_0$ varies strongly with aerosol type, with the highest values of almost 1.0 for the antarctic, maritime clean, and maritime tropical aerosol types. Since water is almost non-absorbing at the considered wavelengths, the water uptake does hardly change $\omega_0$ if $\omega_0$ is already close to 1.0. The single scattering albedo of the desert type is much lower but it is also virtually independent on the RH as this aerosol type does not take up much water. For the other types, an increase of RH results in an increase of $\omega_0$.

The extinction to mass conversion factor $\eta$, which is plotted in the third row of Fig. 5, is necessary to calculate mass concentrations from extinction coefficient measurements or mass loadings from AOD measurements. An important parameter for $\eta$ is the particle size (e.g., Gasteiger et al., 2011a) with the consequence that the desert aerosol type, which contains the highest fraction of coarse particles of the considered types, shows the highest $\eta$ values. Again, the wavelength dependency is significant for the other aerosol types so that the $\eta$ values at $\lambda = 1064$ nm (right column) are significantly larger than at $\lambda = 532$





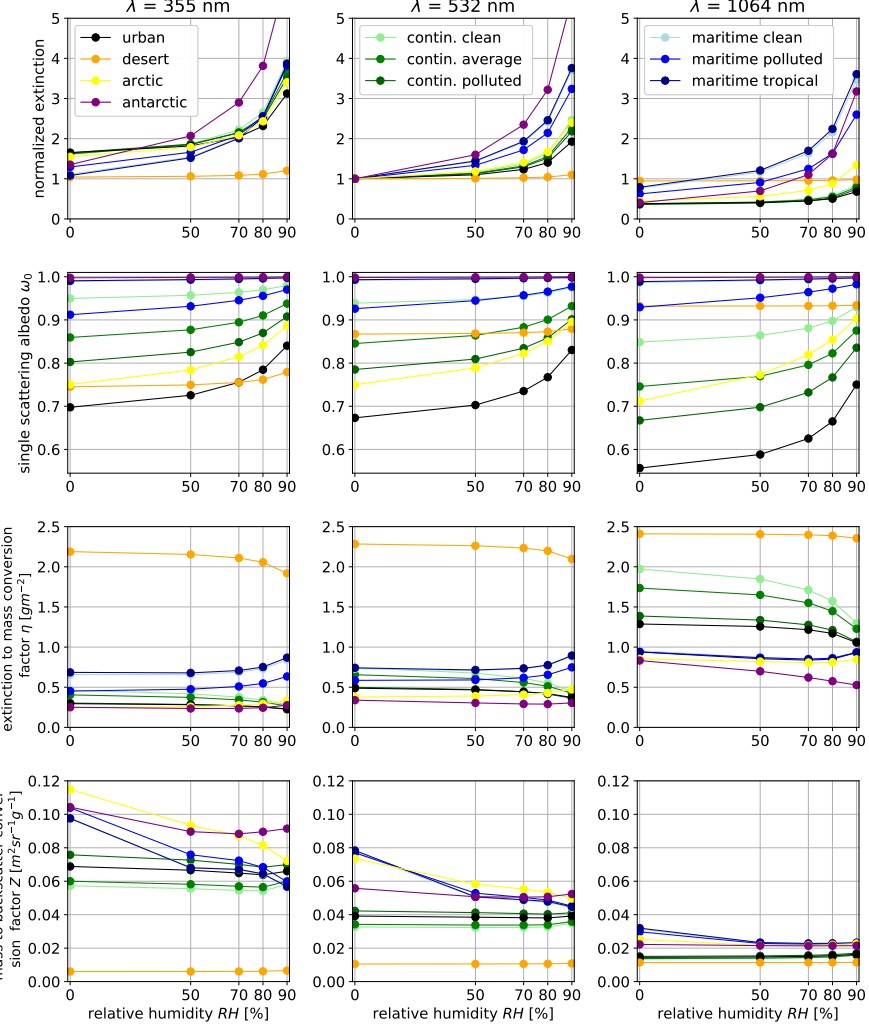

**Figure 5.** Properties of OPAC aerosol types as function of relative humidity RH calculated with the $\kappa$-parameterization (Zieger et al., 2013) implemented in MOPSMAP (Eq. 19). The different colors denote the different OPAC aerosol types as indicated in the legends. The columns denote different wavelengths $\lambda$ as indicated above the upper row. The upper row shows the extinction coefficient normalized to the extinction coefficient of the same aerosol type at RH=0% and $\lambda$=532 nm. The single scattering albedo $\omega_0$, the extinction to mass conversion factor $\eta$, and the mass to backscatter conversion factor $Z$ are plotted in the subsequent rows.

nm (middle column). The dependence of $\eta$ on RH is significantly weaker than the dependence of the extinction on RH (upper row), which may be explained with the compensation of the increase of extinction by the increase of mass with increasing RH.

The bottom row of Fig. 5 illustrates the mass to backscatter conversion factor $Z$ as function of the relative humidity RH. $Z$ is useful for example for comparisons of vertical profiles simulated with aerosol transport models to profiles measured with





lidar or ceilometer. Multiplication of simulated aerosol mass concentrations $M$ with $Z$ provides simulated $\beta$ profiles which can be compared with the measurements. The figure shows that there is considerable spread between the different aerosol types in particular at short wavelengths. RH has strong effects only on the maritime and arctic aerosol types.

Currently the hygroscopic growth of different aerosol components is not ultimately understood, and different $\kappa$-values are discussed. With MOPSMAP their influence on the optical properties can easily be determined and used in validation studies.

## 5.2 Optical properties for sectional aerosol models

Aerosol transport models in combination with the optical properties of the aerosol allow one to model the radiative effect of the aerosol. The aerosol is typically modeled in terms of mass concentrations for a limited number of aerosol types divided over a few size bins (sectional aerosol model) or a few modes (modal aerosol models). Thus, realistic optical properties for each size bin of each aerosol type are required for modeling the radiative effects (e.g., Curci et al., 2015).

In this example, we calculated the optical properties of dust at $\lambda$=500nm for the five size bins of the COSMO-MUSCAT model (Heinold et al., 2007). The size bins are determined by the radius limits $0.1\mu$m, $0.3\mu$m, $0.9\mu$m, $2.6\mu$m, $8\mu$m, and $24\mu$m. We assumed constant $dv/dlnr$ within each bin. Each bin was modeled through the expert mode of the MOPSMAP web interface. The refractive index is $m = 1.53 + 0.0078i$ which is equal to the value given for the mineral components in OPAC. We considered two cases for the particle shape, on the one hand spherical particles and on the other hand prolate spheroids with the aspect ratio distribution given by Kandler et al. (2009). For the latter case we assumed volume-equivalent sizes to keep the particle mass constant.

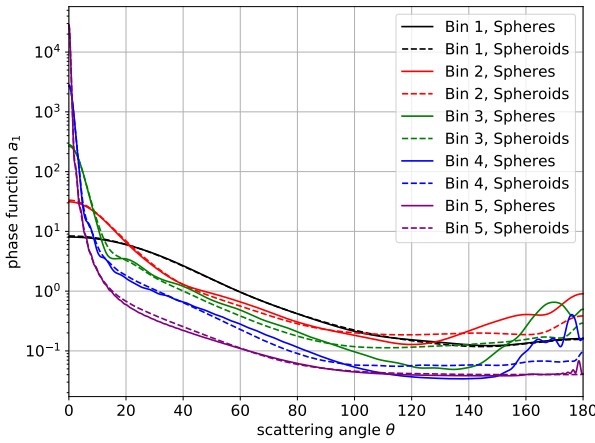

**Figure 6.** Phase functions at $\lambda$=500nm of the five COSMO-MUSCAT dust size bins (different colors) assuming spherical particles (solid lines) and prolate spheroids (dashed lines). For details see text.





**Table 4.** Optical properties at $\lambda$=500nm of the five COSMO-MUSCAT dust size bins. Two cases for the particle shape are considered: Spheres / prolate spheroids. For details see text.

|  | bin 1 | bin 2 | bin 3 | bin 4 | bin 5 |
|---|---|---|---|---|---|
| $\omega_0$ | 0.9632 / 0.9628 | 0.9216 / 0.9264 | 0.7903 / 0.7934 | 0.6450 / 0.6485 | 0.5561 / 0.5601 |
| $g$ | 0.6567 / 0.6585 | 0.6866 / 0.7111 | 0.8088 / 0.8109 | 0.8998 / 0.9017 | 0.9442 / 0.9419 |
| $\eta\,[gm^{-2}]$ | 0.2905 / 0.3000 | 0.5594 / 0.5236 | 2.230 / 2.071 | 6.989 / 6.633 | 22.09 / 20.90 |
| $Z\,[m^2 sr^{-1} g^{-1}]$ | $4.234{\cdot}10^{-2}$ / | $1.185{\cdot}10^{-1}$ / | $1.403{\cdot}10^{-2}$ / | $1.204{\cdot}10^{-3}$ / | $8.225{\cdot}10^{-5}$ / |
|  | $3.981{\cdot}10^{-2}$ | $5.421{\cdot}10^{-2}$ | $8.901{\cdot}10^{-3}$ | $7.457{\cdot}10^{-4}$ | $8.651{\cdot}10^{-5}$ |

The calculated phase functions are presented in Fig. 6, where each size bin is represented by an individual color. The difference between both lines of same color represents the shape effect. For size bin 1 ($0.1\mu$m $< r < 0.3\mu$m, black lines) the difference is small, while the shape effect is larger for all other bins. The strongest effects are found for $\theta > 100°$ with differences of up to a factor of 4 between the particle shapes. These angular ranges can be important for example for the backscattering of sun light into the space and thus for the aerosol radiative effect. The very strong effect at $\theta$=180° is relevant for any lidar application, e.g, the intercomparison of modeled and measured attenuated backscatter profiles (Chan et al., 2017).

Calculated parameters relevant for radiative transfer and remote sensing are given in Tab. 4. The shape effect on the single scattering albedo $\omega_0$ and the asymmetry parameter $g$ is small except for size bin 2 where $g$ is significantly larger for the spheroids than for the spheres. The extinction to mass conversion factor $\eta$ is systematically smaller for spheroids than for spheres in bins 2-5 because the geometric cross section of the spheroids is $\approx 5.5\%$ larger than the cross section of the volume-equivalent spheres. The mass to backscatter conversion factor $Z$ of the spheroids is for most size bins lower than $Z$ of spheres with maximum differences being larger than a factor of 2.

### 5.3 Effect of cut-off at maximum size

Many in-situ measurement setups are limited with respect to the maximum particle size they are able to sample, e.g., because of losses at the inlet or the tubing. In this example, we illustrate the effect of the cut-off for the desert aerosol type from OPAC at $RH$=0%.

Fig. 7 illustrates various aerosol properties as a function of the cutoff radius $r_{max}$. The upper panel shows properties that are normalized by the values found at $r_{max}$=60$\mu$m (where 99.988% of the total particle cross section is covered, referring to $r_{max} = \infty$). The PM10 mass, i.e. the mass in the particles with diameter smaller than 10$\mu$m ($r_{max}$=5$\mu$m), and the PM2.5 mass ($r_{max}$=1.25$\mu$m) are standard parameters to quantify pollution (e.g., Querol et al., 2004). In our example, PM10 and PM2.5 contains only 59.5% and 21.6% of the total particle mass, respectively. However, PM10 and PM2.5 measurement setups cover 94.4% and 69.0% of the total cross section area or extinction, respectively. The single scattering albedo in this case of PM2.5 is about 0.035-0.071 higher than for the total aerosol, while the asymmetry parameter is lowered by about 0.02-0.04. As further



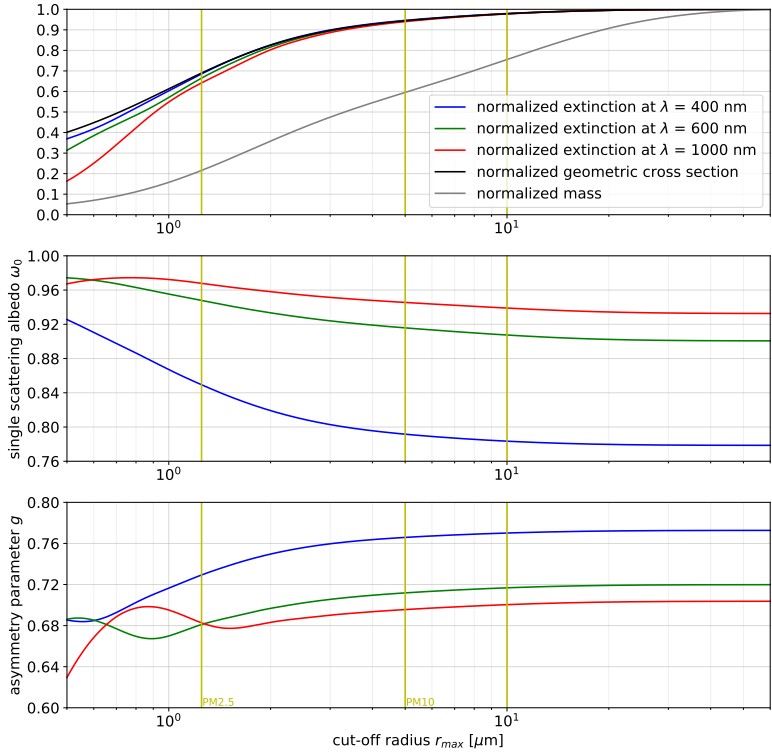

**Figure 7.** Optical and microphysical properties of the OPAC desert aerosol type as function of cutoff radius $r_{max}$. The upper panel shows the normalized extinction at three wavelengths, the normalized cross section area, and the normalized mass. Normalization to values calculated for $r_{max}$=60$\mu$m. The single scattering albedo at the same wavelengths is plotted in middle panel, while the asymmetry parameter is in the lower panel.

example, if the cutoff is $r_{max}$=10$\mu$m, 97.8% of the total cross section and 75.6% of the mass are covered; the single scattering albedo and the asymmetry parameter deviate from the total aerosol by less than 0.008.

This example shows that consideration of maximum size is essential when derived optical properties or mass concentrations are interpreted and results can be severely misleading if the cut-off radius is not considered. These effects can be easily quantified with MOPSMAP and its web interface.



**Table 5.** Properties of one-modal size distribution at $\lambda = 532nm$ consisting of spheres or aggregate particles (shape D, Fig. 1 of Gasteiger et al. (2011b)) assuming different size equivalences. For details see text.

| properties | spheres | aggregate particles | | |
|---|---|---|---|---|
| | using $r$ | using $r_c$ | using $r_v$ | using $r_{vcr}$ |
| $\alpha_{ext}$ [km$^{-1}$] | 0.350 | 0.347 | 0.449 | 0.750 |
| $\omega_0$ | 0.897 | 0.922 | 0.910 | 0.883 |
| $g$ | 0.722 | 0.679 | 0.680 | 0.689 |
| $a_1(0°)$ | 100 | 97.4 | 128 | 222 |
| $\widetilde{a}_1(0°)$ [km$^{-1}$ sr$^{-1}$] | 2.51 | 2.48 | 4.17 | 11.7 |
| $a_1(180°)$ | 1.21 | 0.405 | 0.420 | 0.432 |
| $S$ [sr] | 11.6 | 33.6 | 32.8 | 33.0 |
| $\delta_l$ | 0.000 | 0.450 | 0.454 | 0.454 |
| cross section density $a$ [km$^{-1}$] | 0.141 | 0.141 | 0.186 | 0.323 |
| mass concentration $M$ [$\mu g \cdot m^{-3}$] | 482 | 318 | 481 | 1103 |

## 5.4 Effect of selection of size equivalence of non-spherical particles

In case of a non-spherical particle, the particle size is usually described by the size of an equivalent sphere. As introduced in Sect. 2.1, this can be, for example, a sphere with the same cross section area, the same volume, or the same ratio between volume and cross section as the non-spherical particle. These size equivalences can be selected in MOPSMAP and its web interface.

This example demonstrates effects arising from this selection. A log-normal size distribution with $r_{mod} = 0.5$ $\mu$m, $\sigma = 2$, $r_{min} = 0.001$ $\mu$m, and $r_{max} = 1.75$ $\mu$m ($r_{eff} = 0.98$ $\mu$m) is assumed. $N_0$ is set to $103.66$ $cm^{-3}$, which results in a concentration of $100$ $cm^{-3}$ in the range from $r_{min}$ to $r_{max}$. Furthermore, the particle mass density is set to $2600$ $kgm^{-3}$, the refractive index is $m = 1.54 + 0.005i$ and the wavelength is $\lambda = 0.532$ $\mu$m. In this example particles larger than the wavelength are optically dominant. Results calculated with MOPSMAP are shown in Tab. 5. The first column shows the optical properties of spherical particles. In the subsequent columns, all particles are assumed to be aggregate particles (shape D) with the same $r_c$ (second column), the same $r_v$ (third column), and the same $r_{vcr}$ (last column) as the spheres in the first column. In other words, this means that $r_{mod}$, $r_{min}$, and $r_{max}$ alternately refer to these different radius definitions.

The results show that the size of non-spherical particles increases (see cross section density $a$ and mass concentration $M$) from assuming $r_c$ over $r_v$ to $r_{vcr}$. The extinction coefficient $\alpha_{ext}$ and the forward volume scattering $\widetilde{a}_1(0°)$ of the non-spherical particles best agree with the spherical counterparts if cross-section-equivalence is assumed. These properties are known to be sensitive to the particle cross section for particles larger than the wavelength. The absorption is in first approximation proportional to the particle volume if absorption is weak. As a consequence, for the single scattering albedo $\omega_0$ both cross



**Table 6.** Elements of the Jacobian matrix, i.e. first partial derivatives, of a dust-like ensemble (see text for details).

|              | $\partial\omega_0$ | $\partial g$ | $\partial S$ |
|--------------|--------|--------|-----------|
| $\partial\epsilon'$ | +0.010 | +0.058 | +48.3 sr  |
| $\partial m_r$ | -0.037 | -0.428 | -360 sr   |
| $\partial m_i$ | -11.0  | +3.69  | +2839 sr  |

section and volume are relevant and dependencies are more complicated than for $\alpha_{ext}$. The single scattering albedo $\omega_0$ of shape D decreases in Tab. 5 from left to right due to the strong increase in particle volume. The selection of the size equivalence has a small effect on the asymmetry parameter $g$, the backward phase function $a_1(180°)$, the lidar ratio $S$, and the linear depolarization ratio $\delta_l$.

5    These results highlight the importance of a thoughtful selection of the size equivalence, considering that the most appropriate size equivalence depends on the concept how the size distribution is measured. For example, if scattering by coarse dust particles is measured and the size is inverted assuming spherical particles, assuming cross-section equivalence in subsequent applications with non-spherical particles seems natural as scattering mainly depends on the particle cross section. MOPSMAP and its web interface provides the flexibility to investigate this topic theoretically.

## 5.5   Uncertainty estimation of calculated optical properties

In general, the knowledge on microphysical properties is limited, thus they are subject to uncertainties. If these uncertainties can be quantified, it is consistent to also quantify the corresponding uncertainties of the optical properties.

In this regard, the sensitivity of a calculated optical property $\zeta$ to changes of a microphysical property $\chi$ is an important aspect that can be expressed by the first partial derivative $\partial\zeta/\partial\chi$. The Jacobian matrix $\mathbf{J}$ is the $M \times N$ matrix containing all first partial derivatives for $M$ optical properties and $N$ microphysical properties. The elements of $\mathbf{J}$ of an aerosol ensemble can be numerically calculated by perturbing the microphysical properties of the ensemble. For demonstration in the following example we perturb $\chi$ with a factor of 0.99 and 1.01 to numerically calculate the first partial derivatives. A sample script for the calculation of $\mathbf{J}$ is provided together with MOPSMAP.

Tab. 6 shows an example of $\mathbf{J}$ for the optical properties $\zeta \in \{\omega_0, g, S\}$ and the microphysical properties $\chi \in \{\epsilon', m_r, m_i\}$. $\mathbf{J}$ was calculated for a simplified dust ensemble described by one log-normal size mode with $r_{mod}$=0.1$\mu$m, $\sigma$=2.6, $r_{min}$=0.001$\mu$m, $r_{max}$=20$\mu$m, a refractive index $m$=1.53+0.0063i, and prolate spheroids with $\epsilon'$=2.0. The wavelength is set to $\lambda$=532nm. This results in $\omega_0$=0.9020, $g$=0.7319, and $S$=69.95sr. These properties are most sensitive to $m_i$ which can be clearly seen from Tab. 6. For example, a change of $m_i$ by 0.001 would result in a change of $\omega_0$ of 0.011. But it also needs to be considered that the absolute values of $m_i$ are usually low, which contributes to the high values for the partial derivatives. Nonetheless, by comparing the derivates of $\omega_0$ it is clear that $m_i$ is by far most important for $\omega_0$. An increase in $\epsilon'$ or $m_i$ increases $g$ and $S$, while

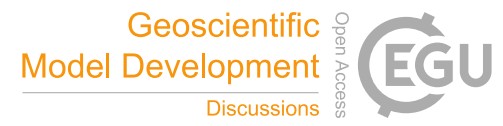

an increase in $m_r$ reduces their values. The sensitivity to perturbations of the microphysical properties is particularly strong for the lidar ratio $S$ which can be seen by comparing $S$=69.95sr of the ensemble with the partial derivatives. We emphasize that the accuracy of $\mathbf{J}$ is limited by the sampling in the MOPSMAP data set (see also Sect. 3.3), for example partial derivatives $\partial \zeta / \partial m_r$ are constant between the $m_r$ grid points of the data set.

The Jacobian matrix $\mathbf{J}$ is valid for a certain set of microphysical properties values and, as mentioned, $\mathbf{J}$ can be used to quantify the uncertainty of the calculated properties for a given microphysical uncertainty. However, when uncertainties of the microphysical properties get larger, $\mathbf{J}$ may change significantly within the uncertainty range of $\chi$ and other approaches may be required to estimate the uncertainty of the calculated optical properties. A simple approach applicable to this problem is the Monte Carlo method. Repeated calculations with microphysical properties randomly chosen within the uncertainty range are performed. The uncertainty of the calculated quantities is determined by the statistics over the different sampled

ensembles. In general, the computation time is longer than using $\mathbf{J}$ and is proportional to the number of calculated ensembles. Due to the statistical nature of the Monte Carlo method, the final results get more precise with increasing number of sampled ensembles. A Monte Carlo script for the uncertainty propagation is provided together with MOPSMAP. For example, based on the ensemble described above, sampling within the uncertainty ranges $r_{mod} = 0.1 \pm 0.01 \mu$m, $\sigma = 2.6 \pm 0.1$, $m_r = 1.53 \pm 0.03$,

$m_r = 0.0063 \pm 0.002$, and $\epsilon' = 2.0 \pm 0.5$ results in the ranges $0.85 < \omega_0 < 0.94$, $0.68 < g < 0.78$, and $29sr < S < 103sr$.

### 5.6    Effect of refractive index variability

Mineral dust aerosols are ensembles of different minerals having different refractive indices. Usually the variability of the refractive index of the particles within a dust aerosol ensemble is neglected when modeling its optical properties. In this example, we compare optical properties calculated using the full measured variability of the imaginary part of the refractive index $m_i$

to properties calculated with the common assumption of all particles in an ensemble having an average $m_i$. Furthermore, a parameterization of the variability is considered.

We use the desert aerosol type of OPAC (Koepke et al., 2015). Prolate spheroids with the aspect ratio distribution of Kandler et al. (2009) are assumed for the mineral components and spherical particles for the WASO component (RH=0%). The real part of the refractive index is $m_r$=1.53 for all particles. The wavelength in this example is set to $\lambda = 355$ nm, which is a wavelength

where absorption by iron oxide is strong. Because of the variable iron oxide content of individual particles, the variability of $m_i$ is large at this wavelength. Consequently, a significant influence on optical properties can be expected. In this example we consider three cases of imaginary part variability: First, we apply the size-resolved distribution of the imaginary part of the refractive index for Saharan dust as derived from mineralogical analysis (Kandler et al., 2011). Second, we assume the average imaginary part for all particles (it is 0.0175 which is close to 0.0166 given for the mineral components in OPAC at $\lambda = 355$

nm). Finally, we parameterize the $m_i$ distribution with the non-absorbing fraction approach as introduced in Sect. 3.1. In this case we set $\mathcal{X}$=0.5, resulting in 50% of the mineral particles having $m_i = 0$, while the other 50% of the particles having $m_i = 0.0349$.





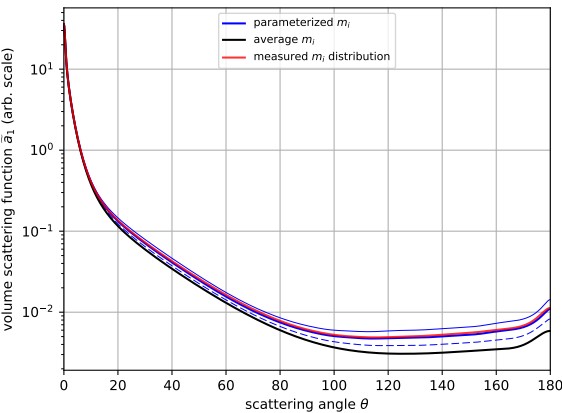

**Figure 8.** Volume scattering function of dust at $\lambda = 355nm$ (arbitrary scale) for different $m_i$ distributions (see text).

Fig. 8 shows the volume scattering function for the three cases. This figure shows that the sensitivity of the forward scattering to the $m_i$ distribution is negligible whereas the sensitivity increases with increasing scattering angle $\theta$. For backward scattering, the difference between the measured $m_i$ distribution (red line) and using the average $m_i$ (black line) is more than a factor of two. The parameterization assuming $\mathcal{X}$=50% (thick blue line) is in much better agreement with the measured case. The root-mean-square relative deviation between the volume scattering function for the measured distribution and for the average $m_i$ is 30%, whereas it is only 4% for the parameterization. For comparison also two additional $\mathcal{X}$ values, i.e. $\mathcal{X}$=25% (thin dashed blue line) as well as $\mathcal{X}$=75% (thin solid blue line), are shown in Fig. 8, but their deviation is larger than for the parameterization with $\mathcal{X}$=50%. The extinction coefficient $\alpha_{ext}$ changes only by less than 0.03% between the three representations of $m_i$. For $\omega_0$ we find 0.852 using the measured $m_i$-distribution, whereas $\omega_0$=0.741 when using the average $m_i$ and $\omega_0$=0.834 using the parameterization with $\mathcal{X}$=50%. For the asymmetry parameter $g$ we find 0.744, 0.789, and 0.749 for the measured, averaged, and parameterized cases, respectively. For the lidar ratio $S$ values of 41sr, 78sr, and 42sr are calculated for the three cases, while for the linear depolarization ratio $\delta_l$ values of 0.241, 0.212, and 0.220 are obtained.

These results emphasize that it is important to consider the non-uniform distribution of the absorptive components in the desert dust ensembles for optical modeling of such aerosols at short wavelengths. We have shown in this example that optical properties of Saharan dust can be well simulated with $\mathcal{X}$=0.5. Whether this conclusion holds for other cases of desert dust can easily be investigated by means of MOPSMAP when measurements of $m_i$ distributions of further dust types are available.

### 5.7 Effect of particle shape on the nephelometer truncation error

Integrating nephelometers aim to measure in situ the total scattering coefficient $\alpha_{sca}^{true}$ of aerosol particles by detecting all scattered light. The angular sensitivities of real nephelometers however deviate from the ideal sensitivity which is the sine of



scattering angle $\theta$. For example nearly-forward or nearly-backward scattered light does not reach the detectors because of the instrument geometry (Müller et al., 2011). This has to be considered during the evaluation of measurements and can be done by applying a truncation correction factor $C_{ts} = \alpha_{sca}^{true}/\alpha_{sca}^{meas}$ to the measured scattering coefficients $\alpha_{sca}^{meas}$. $C_{ts}$ can be calculated theoretically using optical modeling if aerosol microphysical properties and the angular sensitivity of the instrument are known.

Some nephelometers not only measure the total scattering coefficient but also the hemispheric backscattering coefficient which is the scattering integrated from $\theta = 90°$ to $180°$. Also for the hemispheric backscattering coefficient a correction factor needs to be applied to correct the measured hemispheric backscattering coefficient affected by the non-ideal instrument sensitivity. This correction factor $C_{bs}$ is defined analogously to $C_{ts}$ as the ratio between the true coefficient and the measured one. Note that this hemispheric backscattering coefficient is defined different than $\beta$ which is measured by lidars and used elsewhere in

this paper.

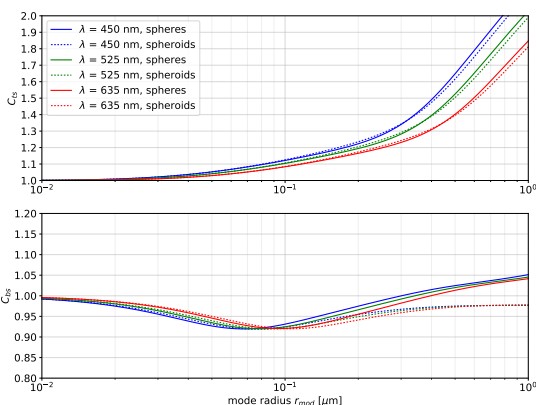

**Figure 9.** Modelled correction factors $C_{ts}$ for total scattering (upper plot) and $C_{bs}$ for hemispheric backscattering (lower plot) of a Aurora 3000 nephelometer as function of particle size. For details see text.

Fig. 9 shows modeled correction factors for the total (upper panel) and the backscatter (lower panel) channel of an Aurora 3000 nephelometer. The angular sensitivity of the instrument is taken from Müller et al. (2011). For the following sensitivity study the mineral dust refractive index from OPAC (Hess et al., 1998), the parameterized $m_i$ distribution with $\mathcal{X}=50\%$ (as shown in Sect. 5.6), a log-normal size mode with $\sigma = 1.6$ and a maximum radius of $r_{max} = 5$ $\mu$m (corresponding to a PM10

inlet) is assumed. The mode radius $r_{mod}$ is varied from 0.01 to 1 $\mu$m (horizontal axis) and two cases for the particle shape, i.e. spherical particles (solid lines) and cross-section-equivalent prolate spheroids with the $\epsilon'$ distribution from Kandler et al. (2009) (dashed lines) are considered. The colors denote the three operating wavelengths of the instrument (450, 525, and 635 nm). The figure shows that the total scattering correction factor $C_{ts}$ mainly depends on particle size. In case of large particles ($r_{mod} = 1$ $\mu$m), the nephelometer underestimates total scattering by a factor of $\approx 2$ if the truncation error is not corrected.

Shape has only a small effect on forward scattering, thus its influence on the correction of the truncation error is less than



3% (compare dashed and solid lines of same color). The maximum shape effect on $C_{bs}$ is 7%, i.e. indicating that assuming spherical particles for the truncation correction may result in an overestimation of the hemispheric backscattering coefficient.

The correction factors might be recalculated for example when new data on the refractive index or particle shape become available. This example highlights the potential of MOPSMAP as a useful tool for the characterization of optical in-situ

instruments. In addition, it could be used for the interpretation of angular measurements, for example as performed with a polar photometer by Horvath et al. (2006).

## 5.8   Optical properties of ash from different volcanoes close to the source

Vogel et al. (2017) present a data set comprising shape-size distributions of ashes from nine different volcanoes as well as wavelength-dependent refractive indices for five different ash types. The particles were collected between 5 and 265 km from

the volcanoes. While refractive indices can be expected to be valid also at larger distances from the volcanoes, the effective radii in the range from $9.5\mu$m to $21\mu$m are probably not realistic for long-range transported ash. Based on this data set, which is available in the supporting information of Vogel et al. (2017), we calculate optical properties of these volcanic ashes with MOPSMAP. Each single particle is modelled as a prolate spheroid with the given size and aspect ratio, as well as with the refractive index given for the type of ash the volcano emits. In addition, we assume a non-absorbing fraction of $\mathcal{X}$=50% (as

used in Sect. 5.6). The application of this non-absorbing fraction approach seems reasonable when taking into account the variability of the transparency of the particles shown in Fig. 5 of Vogel et al. (2017). Due to the data set limits of MOPSMAP, particles with $r > 47.5\ \mu$m are modeled as $r = 47.5\ \mu$m and aspect ratios $> 5$ are set to 5. For each volcano, less than 0.5% of the particles were affected by these modifications.

Fig. 10 shows the single scattering albedo $\omega_0$ and the asymmetry parameter $g$ for the nine ashes as function of wavelength

between 300 nm and 1500 nm. Differences of $\omega_0$ are up to about 0.12 with ash form Chaiten (Chile) and Mt. Kelud (Indonesia) being the least and most absorbing species, respectively. $\omega_0$ is correlated with the ash type, which is mainly a result of the significant variability of $m_i$ (see Fig. 16b of Vogel et al. (2017)). For all ashes, $\omega_0$ increases slightly with wavelength, typically by about 0.05 over the wavelength range shown. The variability of $g$ is less than 0.05 and for all ashes the changes with wavelength are weak with values of less than 0.02. The mass to backscatter conversion factor $Z$ varies between 1.16 to

$3.38 \cdot 10^{-3} m^2 sr^{-1} g^{-1}$ for the nine ashes. The extinction to mass conversion factor $\eta$ at $\lambda$=550nm ranges from 14.8 to 33.0 $gm^{-2}$ which is considerably higher than known for typical aerosols (e.g., Fig. 5) or volcanic ash transported over continental scales (e.g., $\eta$ between 1.10 and 1.88 $gm^{-2}$ found by Wiegner et al. (2012)). In particular the different values of $\eta$ clearly demonstrate that optical properties of volcanic ash layers drastically change with the distance from the eruption due to changing microphysics.

This example suggests that it is worthwhile to consider the specific microphysical properties of each volcano. However, for realistic MOPSMAP calculations valid in the long-range regime certainly size distributions different from the ones used in this example must be applied whereas the refractive indices are more likely representative.





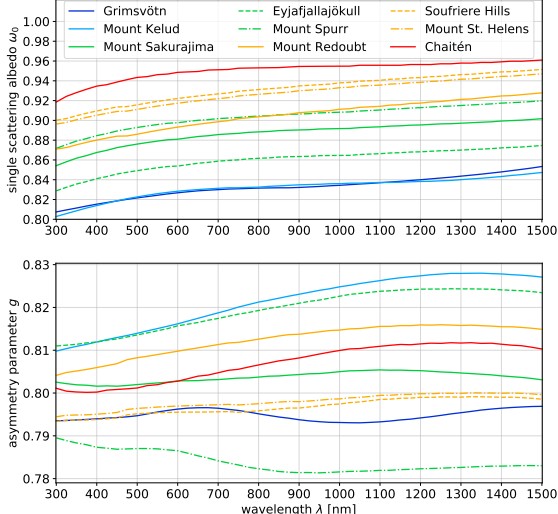

**Figure 10.** Modelled wavelength-dependent optical properties for ashes from different volcanoes. More details on the ash samples is given in Tab. 1 of Vogel et al. (2017). The colors indicate ash type: basalt in dark blue, basaltic andesite in light blue, andesite in green, dacite in orange, and rhyolite in red (see Fig. 7 of Vogel et al. (2017) for reference).

## 6   Conclusions

Radiative properties of atmospheric aerosols are relevant for a wide range of meteorological applications, in particular for radiative transfer calculations and remote sensing and in-situ techniques. Optical properties strongly depend on the microphysical properties of the particles – size, refractive index and shape, properties that are highly variable under ambient conditions. As a consequence, the application of mean properties could be questionable. However, the determination of optical properties of specific aerosol ensembles can be quite time consuming, in particular when non-spherical particles shall be considered.

For this purpose we have developed the MOPSMAP package that provides the full set of optical properties of arbitrary randomly-oriented aerosol ensembles: single particles of the ensemble can be spherical or spheroidal with size parameters up to $x \approx 1000$. Moreover, a small set of irregular particles is considered. The refractive index can be $0.1 \leq m_r \leq 3.0$ and $0 \leq m_i \leq 2.2$. The size distribution of the ensemble can either be parametrized as log-normal distribution, (modified) gamma distribution or freely chosen according to individual data. MOPSMAP includes a web interface for online calculations, offering the most frequently used options; for advanced applications or large sets of computations the full package is available on request. Key applications of MOPSMAP are expected to be the evaluation of radiometer measurements in the UV, VIS and N-IR spectral range, or aerosol lidar measurements. They can help to improve the inversion of such measurements for aerosol




characterization. Furthermore, MOPSMAP can be used to refine optical properties of aerosols in radiative transfer models, or in numerical weather prediction and chemistry transport models.

The details of the concept underlying MOPSMAP are discussed in this paper. Several examples are presented to illustrate the potential of the package, including an example to calculate optical properties for sectional aerosol models and an example

illustrating the effect of maximum size cut-off that occurs in the inlet system of in-situ instruments. In another example, conversion factors between the backscatter coefficient (available from lidar/ceilometer measurements or from numerical forecast models) and the mass concentration of volcanic ashes have been calculated. These conversion factors are relevant to estimate flight safety after volcanic eruptions and vary by about a factor of three between the nine ashes under investigation.

The concept of MOPSMAP allows continuous upgrades to further extend the range of applications. E.g., the resolution of

the refractive index grid can be increased, larger size parameters can be considered, and new sets of irregular particles can be implemented, e.g., those presented by Mehri et al. (2018). However such extensions can be quite time consuming, so that extensions are expected to be limited. Moreover, conceptional upgrades will be investigated without knowing yet whether they can be included in the web interface. Here, a trade-off between scientific complexity and user-friendliness must be found. In this context, the implementation of a core-shell particle model can be discussed. Finally, we want to emphasize that the

feedback of the users will help us to set up a priority list of further actions.

*Code and data availability.* The MOPSMAP data set and the Fortran code are available on request to contact@mopsmap.net.

## Appendix A: Orientation averaging applied to irregular-shaped particles

The particle orientation is specified by three Euler angles ($\alpha_e$, $\beta_e$, $\gamma_e$) as described by Yurkin and Hoekstra (2011). Averaging over $\beta_e$ is done with a step width of 15°, and for each $\beta_e$ up to 24 $\gamma_e$ are used for averaging (see details in Tab. A1). The

averaging over $\alpha_e$ is done within a single ADDA computation because rotation over $\alpha_e$ is equivalent to the rotation of the scattering plane and is computationally cheap. The optical properties are averaged over 32 $\alpha_e$. In total, for a single particle 206 individual ADDA computations are performed and, if the averaging over $\alpha_e$ is considered, 6592 orientations are evaluated. For the numerical calculation of orientation averages of extensive optical properties $\zeta \in \{C_{ext}, C_{sca}, C_{sca} \cdot \mathbf{F}\}$, the following steps were applied.

Averaging over $\alpha_e$ is done by ADDA for each $\beta_e$-$\gamma_e$-pair using the option '-phi_integr 1'. The $\alpha_e$-averaged quantity is denoted as $\zeta_\alpha(\beta_e, \gamma_e)$.



**Table A1.** $\beta_e$ and $\gamma_e$ grid points for orientation averaging of ADDA calculations.

| $\beta_e$ | $\gamma_e$ |
|---|---|
| $0°, 180°$ | $0°$ |
| $15°, 165°$ | $0°, 60°, 120°, ..., 300°$ |
| $30°, 150°$ | $0°, 30°, 60°, ..., 330°$ |
| $45°, 60°, 75°, 90°, 105°, 120°, 135°$ | $0°, 15°, 30°, ..., 345°$ |

As the next step, averaging over $\gamma_e$ is done (outside ADDA) for each $\beta_e$ using

$$\zeta_{\alpha,\gamma}(\beta_e) = \frac{1}{N_\gamma} \sum_{i_\gamma=1}^{N_\gamma} \zeta_\alpha(\beta_e, \gamma_e[i_\gamma]). \tag{A1}$$

where $N_\gamma$ is the number of equidistant $\gamma_e$ grid points for the given $\beta_e$ (Tab. A1).

Finally, averaging over $\beta_e$ is done numerically using linear interpolation of $\zeta_{\alpha,\gamma}$ between the $\beta_e$ grid points and subgrid
5    sampling with $N_s = 100$ subgrid points:

$$\zeta = \sum_{i_\beta=1}^{N_\beta-1} \sum_{i_s=1}^{N_s} \left[ \left( (\zeta_{\alpha,\gamma}(\beta_e[i_\beta+1]) - \zeta_{\alpha,\gamma}(\beta_e[i_\beta])) \frac{i_s}{N_s} + \zeta_{\alpha,\gamma}(\beta_e[i_\beta]) \right) \cdot \right.$$
$$\left. \left( \cos((\beta_e[i_\beta+1] - \beta_e[i_\beta]) \frac{i_s - 0.5}{N_s} + \beta_e[i_\beta]) - \cos((\beta_e[i_\beta+1] - \beta_e[i_\beta]) \frac{i_s + 0.5}{N_s} + \beta_e[i_\beta]) \right) \right] \tag{A2}$$

*Competing interests.* The authors declare that they have no conflict of interest.

*Acknowledgements.* This project has received funding from the European Research Council (ERC) under the European Union's Horizon
10    2020 research and innovation programme (grant agreement No 640458, A-LIFE). The authors thank Michael Mishchenko, Ping Yang, and
Maxim Yurkin for providing their optical modeling codes.




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
