# Peer review of "MOPSMAP v1.0: A versatile tool for modeling of aerosol optical properties"

_Geoscientific Model Development, 2018_

## Short Comment (SC1) · 11 Apr 2018

First, I would like to note that the authors paid a lot of attention to the accuracy of the DDA simulations, and actually quantified it for several representative cases. That is rare among many publications that use the DDA.

Then, I have a couple of suggestion to improve the corresponding discussion. 1) The authors describe the discretization grid for the DDA in terms of the number of number of dipoles per wavelength. But this quantity is not relevant for particles smaller than the wavelength. I guess, the authors used some fixed number of dipoles for smaller particles, but that is not reflected in the text.

[Figure]

2) The orientation-averaging scheme (described in the Appendix A) seems fine, but it is a bit complicated. Thus, it would help if the authors test it for some simple problem (e.g. moderately-sized spheroid), where a reference solution is available. Or at least, mention the results of such tests in the text.

Finally, I wonder if the DDA can be used for cases where neither TMM or IGOM is available (e.g., for 1:3 spheroids with mr<1.04 and size parameters of about 30). The DDA is known to be particularly efficient for such regime (mr close to 1), due to the fast convergence of the (internal) iterative solver. So the authors may at least mention such possibility to extend their dataset.

---

## Referee Comment (RC1) · Anonymous Referee #1 · 24 Apr 2018

The manuscript presents in details a new interesting tool for the modelling of aerosol optical properties. The main innovation is the possibility to use from a web interface, making it potentially appetible for a very wide audience of non-experts in programming. I found the manuscript very clear and well written, with plenty of examples of possible applications that illustrate the potentiality of the code.

I have only a few suggestions to improve the description of the tool:

- page 13: it is described the core of the calculations needed to combine the optical properties of single particles to model particle ensembles, which is key for real-world applications. I found the description from lines 5 to 13 unclear. The authors state

that each aerosol mode is decomposed into eight contributions, stemming from the combination of the interpolation weights on m_r, m_i and epsilon'. I suggest to include a simple example, perhaps with only two variables (e.g. m_r and m_i), to allow a better understanding of this algorithm with summation over weights combinations.

- page 13: in the last paragraph, it is mentioned that the combinations are generally J and not eight, but the reason is unclear. Please clarify with an example.

- page 13: the final calculations of ensemble optical properties are a summation over the aerosol modes included by the user. I suggest to add the information, in this paragraph, that this is equivalent to the so-called "external mixing" assumptions, compared to possible "internal mixing" assumption on the mixing state.

- page 18, line 12: the URL of the tool's web interface is given here for the first time in the manuscript. I suggest to put the information also in the abstract, for more immediacy.

———————————————————

---

## Short Comment (SC2) · 27 Apr 2018

As explained in https://www.geoscientific-model-development.net/about/manuscript_types.html GMD is encouraging authors to upload the program code of the model described in the manuscript (including relevant data sets) as a supplement or make the code and data available at a data repository preferable with an associated DOI (digital object identifier) for the exact model version described in the paper. Access via e-mail request is not considered a persistent access mode and authors need to give reasons in the manuscript why access is restricted. Typically this would be for reasons of licensing.

[Figure]

Lutz Gross GMD Executive Editor

---

## Referee Comment (RC2) · Anonymous Referee #2 · 10 May 2018

The manuscript presents a potentially very helpful tool for the aerosol remote sensing community, MOPSMAP v0.9, including some possible applications for the software. The description of the software characteristics is thorough, providing the future user with a clear picture of the software capabilities, along with its "limitations" (e.g. interpolation and sampling error). I strongly support the manuscript publication, after the following (minor corrections).

1. Make the fonts in the Figures/plots large enough for easy visualization.

2. Section 2.1: Provide the definition of the irregular-shape radius.

3. Page 4, lines 27-28, page 5, line 1: "The real part mr determines the speed of light

inside the particle and therefore the refraction of waves on the particle surface in the macroscopic sense": I think this is an over-simplification that may be misleading for a young scientist. It is better to omit it. Otherwise, please provide relevant reference.

4.  Page 7, line 21: "The minimum size parameter was selected depending on the maximum size achieved with TMM.": An evaluation of the agreement between the 2 methods is missing here. Please provide an indicative plot, containing e.g. the scattering matrix elements $\alpha 1$ and $-b1/a1$ (two sub-plots) for indicative cases (e.g. see Fig.2 in Dubovik et al. (2006) -"Application of spheroid models to account for aerosol particle nonsphericity in remote sensing of desert dust")

5.  Page 10, lines 8-9: "The transition size parameter between TMM and IGOM is in the range $5 < x < 125$, strongly depending on m and particle shape.": Provide the corresponding ranges for different m and particle shapes in an Appendix.

6.  Page 13, lines 5-17: "In case of fixed values of . . . for each mode.": Re-write this section in a more clear way, maybe using some examples. It is not clear what your methodology is here.

7.  Page 20, lines 19-22: "For the continental . . . sea salt particles.": Provide relevant references.

8.  Page 25, lines 12-13: "In other words. . . radius definitions": Provide a visualization of this discussion in a plot with size distributions corresponding to the different radius definitions.

9.  Page 26, line 23-24: "But it also needs . . . partial derivatives": It is not clear what you mean here, it is better to omit this.

10.  Page 27, lines 8-13: "A simple approach. . . together with MOPSMAP.": Provide relevant reference(s).

---

## Author Response (AR1)

*Author's response to discussion of "MOPSMAP v0.9: A versatile tool for modeling of aerosol optical properties" by Josef Gasteiger and Matthias Wiegner, gmd-2018-56*

Dear Klaus Gierens,

please find below our responses to the reviews and short comments as well as the updated manuscript with tracked changes since our discussion paper and a new Supplement. The changes in the manuscript are mainly based on the suggestions we got. We decided to add a Supplement where we demonstrate some aspects in more detail than required for the average reader. Furthermore, we integrated a number of smaller improvements (e.g. correct space around "=" or units) and mention that our irregular shape data set now covers size parameters up to 30.2 instead of 27.5.

We also uploaded our Fortran code and data set to https://zenodo.org but did not yet finalize/publish it so that we can do some last modifications if necessary. The code and data set will have a DOI as suggested by Lutz Gross.

Best regards,

Josef Gasteiger and Matthias Wiegner

Authors' response to comments by reviewer 1:

We are grateful to reviewer 1 for his/her positive rating of our manuscript and useful suggestions. In the following, comments by the reviewer are in italic font, our answers in normal font.

Moreover we want to refer to the revised manuscript where all changes can easily be tracked. We decided to not repeat all changes below for reasons of clarity.

*page 13: it is described the core of the calculations needed to combine the optical properties of single particles to model particle ensembles, which is key for real-world applications. I found the description from lines 5 to 13 unclear. The authors state that each aerosol mode is decomposed into eight contributions, stemming from the combination of the interpolation weights on $m_r$, $m_i$ and $\epsilon'$. I suggest to include a simple example, perhaps with only two variables (e.g. $m_r$ and $m_i$), to allow a better understanding of this algorithm with summation over weights combinations.*

We agree that this description needs to be improved and thus rephrased it in the revised manuscript. Furthermore, we added an example in Section S4 of the Supplement and refer to this example in the paper.

*page 13: in the last paragraph, it is mentioned that the combinations are generally J and not eight, but the reason is unclear. Please clarify with an example.*

This confusion is probably also a result of the previous paragraphs. The case mentioned (a fixed m_r, m_i and $\epsilon_m$ given by the user) where eight contributions are necessary is just an example. For example, if an $\epsilon_m$ distribution is given by the user instead of one fixed $\epsilon_m$, a larger number of contributions is required for a mode. Furthermore, if the ensemble consists of more than one mode the contributions from the different modes need to be added, thus $J$ increases as well. We rearranged and improved this section such that this point should be more clear now. Moreover, Section S4 of the Supplement will certainly help to make clear how the interpolation is done (in an "easy to follow case").

*page 13: the final calculations of ensemble optical properties are a summation over the aerosol modes included by the user. I suggest to add the information, in this paragraph, that this is equivalent to the so-called "external mixing" assumptions, compared to possible "internal mixing" assumption on the mixing state.*

The reviewer's suggestion helps the reader to understand better the context, so we added a sentence 'This approach corresponds to the so-called external mixing of particles.' in this paragraph.

*page 18, line 12: the URL of the tools web interface is given here for the first time in the manuscript. I suggest to put the information also in the abstract,*

*for more immediacy.*

We agree that the URL should be given in the abstract and changed it accordingly. Moreover, we mention the URL also at the end of the introduction and in the conclusions.

Authors' response to comments by reviewer 2:

We appreciate the positive assessment and the useful comments of reviewer 2. In the following, comments by the reviewer are in italic font, our answers in normal font.

Moreover we want to refer to the revised manuscript where all changes can easily be tracked. We decided to not repeat all changes below for reasons of clarity.

*1. Make the fonts in the Figures/plots large enough for easy visualization.*

We agree that the fonts of several figures were too small. We went through the manuscript and improved the figures' font sizes where required for better visualization.

*2. Section 2.1: Provide the definition of the irregular-shape radius.*

Eqs. 1, 2, and 4 are valid also for irregular particles. Maybe this was a bit unclear because the previous paragraph concerns mainly spheroids. We now mention explicitly that the size definitions are valid for 'any kind of non-spherical particles'. This was included before the above mentioned equations.

*3. Page 4, lines 27-28, page 5, line 1: The real part mr determines the speed of light inside the particle and therefore the refraction of waves on the particle surface in the macroscopic sense: I think this is an over-simplification that may be misleading for a young scientist. It is better to omit it. Otherwise, please provide relevant reference.*

As this part is not essential for the paper we removed it as suggested.

*4. Page 7, line 21: The minimum size parameter was selected depending on the maximum size achieved with TMM.: An evaluation of the agreement between the 2 methods is missing here. Please provide an indicative plot, containing e.g. the scattering matrix elements $\alpha_1$ and $-b_1/a_1$ (two sub-plots) for indicative cases (e.g. see Fig.2 in Dubovik et al. (2006) -Application of spheroid models to account for aerosol particle nonsphericity in remote sensing of desert dust)*

We have added several plots similar to Fig. 2 of Dubovik et al. (2006) as Section S3 to the supplement. They illustrate the transition from TMM to IGOM which be briefly discuss. In Section 2.3 of the paper we refer to this Supplement.

*5. Page 10, lines 8-9: The transition size parameter between TMM and IGOM is in the range $5 < x < 125$, strongly depending on m and particle shape.: Provide the corresponding ranges for different m and particle shapes in an Appendix.*

We have added an overview table as Section S2 of the Supplement and in addition have uploaded to https://zenodo.org a detailed list with maximum size parameters of TMM for all 22680 combinations of refractive indices and shapes

included in our MOPSMAP data set of spheroids.

*6. Page 13, lines 5-17: In case of fixed values of . . . for each mode.: Re-write this section in a more clear way, maybe using some examples. It is not clear what your methodology is here.*

Reviewer 1 also had a similar concern. We have rewritten this part and also included an example as Section S4 of the Supplement.

*7. Page 20, lines 19-22: For the continental . . . sea salt particles.: Provide relevant references.*

We added relevant references (Petters and Kreidenweis, 2007; Markelj et al., 2017; Enroth et al., 2018; Psichoudaki et al., 2018) for the $\kappa$ values of different aerosol types.

*8. Page 25, lines 12-13: In other words. . . radius definitions: Provide a visualization of this discussion in a plot with size distributions corresponding to the different radius definitions.*

We added a new Figure 8 to clarify how the selected radius definition affects the results. The corresponding explanations in Sect. 5.4 were rephrased and extended.

*9. Page 26, line 23-24: But it also needs . . . partial derivatives: It is not clear what you mean here, it is better to omit this.*

We agree that this part is not essential for the understanding of this section. So we removed it as suggested.

*10. Page 27, lines 8-13: A simple approach. . . together with MOPSMAP.: Provide relevant reference(s).*

We added the following reference where the Monte Carlo error propagation is discussed: 'Evaluation of measurement data - Supplement 1 to the "Guide to the expression of uncertainty in measurement" - Propagation of distributions using a Monte Carlo method', Tech. rep., Joint Committee for Guides in Metrology, https://www.bipm.org/en/publications/guides/gum.html, 2008.

Authors' response to short comment by Maxim Yurkin:

We appreciate the short comment by Maxim Yurkin with his positive judgement and helpful suggestions. In the following, comments by Maxim Yurkin are in italic font, our answers in normal font.

Moreover we want to refer to the revised manuscript where all changes can easily be tracked. Sect. 2.2.4 of the manuscript and Sect. S1 of the new Supplement are attached to this reply.

*1) The authors describe the discretization grid for the DDA in terms of the number of number of dipoles per wavelength. But this quantity is not relevant for particles smaller than the wavelength. I guess, the authors used some fixed number of dipoles for smaller particles, but that is not reflected in the text.*

Yes, Maxim Yurkin is right, we overlooked this point when writing our discussion paper. It is now included in the ADDA section (2.2.4) of the revised version. We use the dipole set that has 11 dipoles per wavelength at $x = 10$ (with about 23000 dipoles) also for size parameters $x < 10$.

*2) The orientation-averaging scheme (described in the Appendix A) seems fine, but it is a bit complicated. Thus, it would help if the authors test it for some simple problem (e.g. moderately-sized spheroid), where a reference solution is available. Or at least, mention the results of such tests in the text.*

We agree that the description of the orientation-averaging scheme is a bit complicated in the discussion paper, mainly Eq. A2. In the revised paper we replaced Eq. A2 by a more simple equation describing the same method. We feel that it is not necessary to further simplify the scheme as the idea behind is straightforward. Note, that we moved the Appendix of the discussion paper to the Supplement (Section S1.1) of the revised version (to extend it with more details of the accuracy test, see below).

We decided to provide more details of the orientation averaging accuracy tests. However, in order to limit the size of the paper, details are swapped to Section S1.2 of the Supplement whereas our main results are still summarized in the paper. Furthermore, we considered a third irregular shape (F) for the orientation averaging accuracy tests.

Following the suggestion of testing our scheme for a simple problem like spheroids, we also added a test with spheroids at size parameter 2, 4, and 10 as Section S1.3 to the Supplement. We applied ADDA together with our orientation averaging scheme (for simplicity without considering the symmetry of spheroids) and compare the orientation-averaged properties of the spheroids to those calculated with TMM. A brief discussion of the agreement is also included in the Supplement.

*3) Finally, I wonder if the DDA can be used for cases where neither TMM or IGOM is available (e.g., for 1:3 spheroids with $m_r < 1.04$ and size parameters*

*of about 30). The DDA is known to be particularly efficient for such regime (mr close to 1), due to the fast convergence of the (internal) iterative solver. So the authors may at least mention such possibility to extend their dataset.*

As we are not very familiar with atmospheric applications in this refractive index range we did not put much effort in maximizing the size coverage for $m_r < 1.04$. However this suggestion points to a useful future extension of the data set, which is now mentioned in the outlook.

Authors' response to short comment by Lutz Gross:

We thank Lutz Gross for his suggestion.

We uploaded the Fortran code including the dataset (about 30 GB compressed) for open access to https://zenodo.org. A DOI is assigned to the MOPSMAP model version v1.0 of the final GMD paper (in GMDD v0.9 was discussed) and a link is added to the code availability section of the paper. Furthermore, the code archive under https://zenodo.org includes a few examples of MOPSMAP applications corresponding to the examples presented in our paper.

[revised manuscript text omitted]

gmd-2018-56

**Contents**

**S1 Orientation averaging of irregularly-shaped particles and accuracy assessment**

**S1.1 Orientation averaging scheme**

The particle orientation is specified by three Euler angles $(\alpha_e, \beta_e, \gamma_e)$ as described by Yurkin and Hoekstra (2011). Averaging over $\beta_e$ is done with a step width of 15°, and for each $\beta_e$ up to 24 $\gamma_e$ are used for averaging (see details in Table S1). The averaging over $\alpha_e$ is done within a single ADDA computation because rotation over $\alpha_e$ is equivalent to the rotation of the scattering plane and is computationally cheap. The optical properties are averaged over 32 $\alpha_e$. In total, for a single particle 206 individual ADDA runs are performed and, if the averaging over $\alpha_e$ is considered, 6592 orientations are evaluated. For the numerical calculation of orientation averages of extensive optical properties $\zeta \in \{C_{ext}, C_{sca}, C_{sca} \cdot \mathbf{F}\}$, the following steps were applied.

Table S1: $\beta_e$ and $\gamma_e$ grid points for irregularly-shaped particles in MOPSMAP data set.

| $\beta_e$ | $\gamma_e$ |
|---|---|
| 0°, 180° | 0° |
| 15°, 165° | 0°, 60°, 120°, ... , 300° |
| 30°, 150° | 0°, 30°, 60°, ... , 330° |
| 45°, 60°, 75°, 90°, 105°, 120°, 135° | 0°, 15°, 30°, ... , 345° |

Averaging over $\alpha_e$ is done by ADDA for each $\beta_e$-$\gamma_e$-pair using the option '-phi_integr 1'. The $\alpha_e$-averaged quantity is denoted as $\zeta_\alpha(\beta_e, \gamma_e)$.

As the next step, averaging over $\gamma_e$ is done (outside ADDA) for each $\beta_e$ using

$$\zeta_{\alpha,\gamma}(\beta_e) = \frac{1}{N_\gamma} \sum_{i_\gamma=1}^{N_\gamma} \zeta_\alpha(\beta_e, \gamma_e[i_\gamma]). \tag{S1}$$

where $N_\gamma$ is the number of equidistant $\gamma_e$ grid points for the given $\beta_e$ (Table S1).

Finally, the orientation-averaged $\zeta$ is obtained by averaging over $\beta_e$ using

$$\zeta = \frac{1}{2} \int_{0°}^{180°} \widetilde{\zeta}_{\alpha,\gamma}(\beta_e) \cdot \sin\beta_e \cdot d\beta_e \tag{S2}$$

where $\widetilde{\zeta}_{\alpha,\gamma}(\beta_e)$ is linearly interpolated between the available $\zeta_{\alpha,\gamma}(\beta_e)$ grid points (Eq. S1, Table S1). Numerical integration of Eq. S2 is performed using a step width of $\Delta\beta_e = 0.15°$.

**S1.2 Accuracy assessment using dense Euler angle grid**

Here we present a comparison of single particle properties calculated either with the orientation averaging scheme used in the MOPSMAP data set (Sect. S1.1) or using a much denser grid of orientation angles (5° step size for $\beta_e$ and $\gamma_e$). This comparison provides an estimation of the accuracy of the former using the latter as reference.

Irregular shapes B, C, and F with $m = 1.52 + 0.0043i$ at six size parameters $x_v$ from 10.0 to 20.8 are considered, i.e., 18 different single particles. In the following, tables are shown for the extinction efficiency $q_{ext}$ (Table S2), scattering efficiency $q_{sca}$ (Table S3), forward scattering phase function $a_1(0°)$ (Table S4), backscattering phase function $a_1(180°)$ (Table S5), and the normalized 2,2-element of the scattering matrix at backward direction $a_2(180°)/a_1(180°)$ (Table S6), where each line corresponds to a size parameter and the three values separated by slashes correspond to the different irregular shapes B/C/F.

In case of $q_{ext}$ and $q_{sca}$, i.e. properties integrated over all scattering angles, the deviation between MOPSMAP and the reference is virtually zero (Tables S2, S3). The same is true for forward scattering (Table S4). Larger deviations typically in the order of a few percent (max. 14%) occur in case of backscattering. It is well known that scattering under 180° is very sensitive to various parameters of a scattering problem, here the particle orientation. The effect on atmospheric aerosols is however reduced as under realistic conditions over- and underestimates partly compensate according to the ensemble of different particles.

Table S2: Extinction efficiency $q_{ext}$ from the MOPSMAP data set compared to results obtained using a dense grid of step size 5° for $\beta_e$ and $\gamma_e$. The three values separated by slashes correspond to shapes B, C, and F. The relative deviation of $q_{ext}$ of MOPSMAP from the reference is rounded to full percent values.

| size parameter | MOPSMAP data set | dense $\beta_e$ and $\gamma_e$ grid | rel. deviation in % |
|---|---|---|---|
| $x_v$=10.0 | 2.269/2.064/2.069 | 2.267/2.053/2.076 | 0/+1/0 |
| $x_v$=12.0 | 2.521/2.442/2.279 | 2.530/2.430/2.075 | 0/+1/0 |
| $x_v$=14.4 | 2.180/2.367/2.272 | 2.184/2.371/2.280 | 0/0/0 |
| $x_v$=17.3 | 2.203/2.097/2.090 | 2.205/2.100/2.101 | 0/0/-1 |
| $x_v$=19.0 | 2.303/2.200/2.106 | 2.306/2.200/2.114 | 0/0/0 |
| $x_v$=20.8 | 2.229/2.289/2.195 | 2.232/2.297/2.181 | 0/0/+1 |

Table S3: Scattering efficiency $q_{sca}$ from the MOPSMAP data set compared to results obtained using a dense grid of step size 5° for $\beta_e$ and $\gamma_e$. The three values separated by slashes correspond to shapes B, C, and F. The relative deviation of $q_{sca}$ of MOPSMAP from the reference is rounded to full percent values.

| size parameter | MOPSMAP data set | dense $\beta_e$ and $\gamma_e$ grid | rel. deviation in % |
|---|---|---|---|
| $x_v$=10.0 | 2.076/1.881/1.904 | 2.074/1.870/1.910 | 0/+1/0 |
| $x_v$=12.0 | 2.295/2.226/2.082 | 2.303/2.214/2.075 | 0/+1/0 |
| $x_v$=14.4 | 1.916/2.114/2.042 | 1.920/2.118/2.049 | 0/0/0 |
| $x_v$=17.3 | 1.898/1.803/1.822 | 1.899/1.807/1.832 | 0/0/-1 |
| $x_v$=19.0 | 1.975/1.885/1.816 | 1.977/1.886/1.824 | 0/0/0 |
| $x_v$=20.8 | 1.879/1.954/1.883 | 1.881/1.961/1.869 | 0/0/+1 |

Table S4: Forward scattering phase function $a_1(0°)$ from the MOPSMAP data set compared to results obtained using a dense grid of step size 5° for $\beta_e$ and $\gamma_e$. The three values separated by slashes correspond to shapes B, C, and F. The relative deviation of $a_1(0°)$ of MOPSMAP from the reference is rounded to full percent values.

| size parameter | MOPSMAP data set | dense $\beta_e$ and $\gamma_e$ grid | rel. deviation in % |
|---|---|---|---|
| $x_v$=10.0 | 74.00/70.20/78.66 | 73.94/69.62/78.62 | 0/+1/0 |
| $x_v$=12.0 | 119.8/121.9/126.7 | 120.3/121.6/126.8 | 0/0/0 |
| $x_v$=14.4 | 154.7/176.6/188.7 | 154.8/176.8/188.8 | 0/0/0 |
| $x_v$=17.3 | 224.8/225.4/253.3 | 224.8/225.5/253.2 | 0/0/0 |
| $x_v$=19.0 | 288.1/288.4/309.4 | 288.0/288.2/309.6 | 0/0/0 |
| $x_v$=20.8 | 340.4/366.8/391.0 | 340.7/367.2/389.5 | 0/0/0 |

Table S5: Backward scattering phase function $a_1(180°)$ from the MOPSMAP data set compared to results obtained using a dense grid of step size 5° for $\beta_e$ and $\gamma_e$. The three values separated by slashes correspond to shapes B, C, and F. The relative deviation of $a_1(180°)$ of MOPSMAP from the reference is rounded to full percent values.

| size parameter | MOPSMAP data set | dense $\beta_e$ and $\gamma_e$ grid | rel. deviation in % |
|---|---|---|---|
| $x_v$=10.0 | 0.3855/0.3277/0.3184 | 0.3758/0.3262/0.3084 | +3/0/+3 |
| $x_v$=12.0 | 0.3445/0.2643/0.3270 | 0.3238/0.2557/0.3254 | +6/+3/+1 |
| $x_v$=14.4 | 0.3488/0.2409/0.4018 | 0.3423/0.2492/0.3844 | +2/-3/+5 |
| $x_v$=17.3 | 0.3244/0.2423/0.4634 | 0.3091/0.2581/0.4306 | +5/-6/+8 |
| $x_v$=19.0 | 0.2866/0.2200/0.4625 | 0.2702/0.2174/0.4115 | +6/+1/+12 |
| $x_v$=20.8 | 0.2882/0.2185/0.4352 | 0.2800/0.2170/0.3822 | +3/+1/+14 |

Table S6: Normalized (2,2)-element of the scattering matrix at backward direction $a_2(180°)/a_1(180°)$ from the MOPSMAP data set compared to results obtained using a dense grid of step size 5° for $\beta_e$ and $\gamma_e$. The three values separated by slashes correspond to shapes B, C, and F. The relative deviation of $a_2(180°)/a_1(180°)$ of MOPSMAP from the reference is rounded to full percent values.

| size parameter | MOPSMAP data set | dense $\beta_e$ and $\gamma_e$ grid | rel. deviation in % |
|---|---|---|---|
| $x_v$=10.0 | 0.4894/0.5856/0.3342 | 0.5081/0.5166/0.3347 | -4/+13/0 |
| $x_v$=12.0 | 0.5398/0.5713/0.3195 | 0.5135/0.5407/0.3417 | +5/+6/-6 |
| $x_v$=14.4 | 0.5356/0.5489/0.3229 | 0.5200/0.5385/0.3396 | +3/+2/-5 |
| $x_v$=17.3 | 0.4884/0.5684/0.3274 | 0.4890/0.5454/0.3574 | 0/+4/-8 |
| $x_v$=19.0 | 0.4799/0.5362/0.3343 | 0.4800/0.5159/0.3606 | 0/+4/-7 |
| $x_v$=20.8 | 0.5188/0.5382/0.3294 | 0.4973/0.5387/0.3590 | +4/0/-8 |

**S1.3 Accuracy assessment using spheroids**

Here ADDA together with the orientation averaging scheme used for the irregular shapes in the MOPSMAP data set (as described in Sect. S1.1) is applied to prolate spheroids with $m = 1.52 + 0.0043i$ and $\epsilon'$=2.0 (ADDA option '-shape ellipsoid 1.0 2.0'). Volume-equivalent size parameters $x_v$=2, 4, and 10 are considered. As reference the optical properties of the same randomly-oriented particles are calculated with the TMM code of Mishchenko and Travis (1998).

The comparison again shows that the integrated parameters and the forward scattering almost perfectly agree whereas for backscattering few cases with larger deviations up to 17 % are obtained. In general, the relative deviations are of similar magnitude as those found in our tests of Sect. S1.2 though the number of independent ADDA calculations is lower for spheroids than for irregular particles because of the rotation symmetry of spheroids.

Table S7: Properties of prolate spheroids with $\epsilon'$=2.0 and $m = 1.52 + 0.0043i$. The three values separated by slashes correspond to size parameters $x_v$=2, 4, and 10. The relative deviation of ADDA from TMM is rounded to full percent values.

| optical parameter | ADDA + orient. avg. | TMM | rel. deviation in % |
|---|---|---|---|
| $q_{ext}$ | 1.656/3.861/2.274 | 1.650/3.860/2.256 | 0/0/+1 |
| $q_{sca}$ | 1.621/3.780/2.068 | 1.615/3.778/2.048 | 0/0/+1 |
| $a_1(0°)$ | 5.741/18.86/69.30 | 5.748/18.84/69.20 | 0/0/0 |
| $a_1(180°)$ | 0.1069/0.1700/0.3597 | 0.0948/0.1686/0.3568 | +13/+1/+1 |
| $a_2(180°)/a_1(180°)$ | 0.9350/0.5175/0.4721 | 0.9230/0.4927/0.5678 | +1/+5/-17 |

**S2 Maximum TMM size parameters**

A detailed list of maximum TMM size parameters for all 22680 combinations of refractive index and shape from the MOPSMAP spheroid data set is provided for download together with the data set at `https://doi.org/10.5281/zenodo.1284217`. A summary is given in Table S8 for different aspect ratios $\epsilon'$ and refractive index ranges. The TMM calculations for a given refractive index and shape (always iterating from small to large $x_c$ with steps of 5 %) are terminated either when a size parameter equal or larger than the minimum upper size parameter given in Table S8 is successfully modeled or if TMM does not converge. Furthermore, the results were checked for plausibility as discussed in our paper. As a consequence, the maximum TMM size parameter in the MOPSMAP data set is often 0 % - 5 % larger than given in Table S8 but can be smaller in case of numerical problems for specific shape and refractive index combinations.

Table S8: Minimum upper size parameters $x_c$ for the TMM calculations of oblate and prolate spheroids as function of aspect ratio $\epsilon'$ for different refractive index ranges. For details see text.

|  | $1.28 \leq m_r \leq 1.6$ and $m_i \leq 0.1376$ | $1.64 \leq m_r \leq 2$ and $m_i \leq 0.1376$ | all other $m$ with $m_r \geq 1.0$ | $m_r < 1$ |
|---|---|---|---|---|
| $\epsilon' = 1.2$ | 120 | 118 | 60 | 25 |
| $\epsilon' = 1.4$ | 120 | 84 | 60 | 25 |
| $\epsilon' = 1.6$ | 110 | 76.2 | 55 | 25 |
| $\epsilon' = 1.8$ | 88 | 65.8 | 44 | 25 |
| $\epsilon' = 2.0$ | 63 | 56.8 | 31.5 | 25 |
| $\epsilon' = 2.2$ | 51 | 44.5 | 25.5 | 10 |
| $\epsilon' = 2.4$ | 40 | 31.6 | 20.0 | 10 |
| $\epsilon' = 2.6$ | 33 | 24.8 | 16.5 | 10 |
| $\epsilon' = 2.8$ | 29 | 21.4 | 14.5 | 10 |
| $\epsilon' = 3.0$ | 25 | 20.4 | 12.5 | 10 |
| $\epsilon' = 3.4$ | 19.4 | 16.7 | 9.7 | 5 |
| $\epsilon' = 3.8$ | 16.7 | 14.5 | 8.4 | 5 |
| $\epsilon' = 4.2$ | 14.5 | 13.1 | 7.3 | 5 |
| $\epsilon' = 4.6$ | 12.5 | 11.9 | 6.3 | 5 |
| $\epsilon' = 5.0$ | 11.3 | 10.0 | 5.7 | 5 |

**S3 Transition from TMM to IGOM**

Figs. S1 to S8 illustrate the angle-specific scattering as function of size parameter calculated by the TMM (solid lines) and the IGOM approximation (dashed lines). The transitions between TMM and IGOM are shown as dotted lines. The upper plots are similar to those presented in Fig. 2a of Dubovik et al. (2006) (shifted by some factor) but also include results for backscattering (180°). The lower plots show the ratio between the scattering matrix elements $a_2$ and $a_1$. All plots are for $m=1.52+0.0030406i$, close to the value used by Dubovik et al. (2006), and the different plots show different particle shapes.

The figures show a very smooth transition between TMM and IGOM for the scattering intensity in the forward direction. For the backward direction some cases with jumps at the transition size parameters are found, indicating uncertainties of the IGOM approximation for the corresponding scattering angle. It is well known that backscattering is particularly difficult to model. Note that the contribution of IGOM results to the optical properties of typical dust ensembles at visible wavelengths is low so that uncertainties of ensemble backscattering properties arising from IGOM uncertainties are normally limited.

[Figure]

Figure S1: Comparison of IGOM (dotted lines) with TMM (solid lines) for **prolate spheroids with $\epsilon'$=5.0** and $m$=1.52+0.0030406i at different scattering angles $\theta$ (indicated by color).

[Figure]

Figure S2: Comparison of IGOM (dotted lines) with TMM (solid lines) for **prolate spheroids with** $\epsilon'$**=2.8** and $m$=1.52+0.0030406i at different scattering angles $\theta$ (indicated by color).

[Figure]

Figure S3: Comparison of IGOM (dotted lines) with TMM (solid lines) for **prolate spheroids with $\epsilon'=2.0$** and $m=1.52+0.0030406i$ at different scattering angles $\theta$ (indicated by color).

[Figure]

Figure S4: Comparison of IGOM (dotted lines) with TMM (solid lines) for **prolate spheroids with $\epsilon'$=1.4** and $m$=1.52+0.0030406i at different scattering angles $\theta$ (indicated by color).

[Figure]

Figure S5: Comparison of IGOM (dotted lines) with TMM (solid lines) for **oblate spheroids with $\epsilon'$=1.4** and $m$=1.52+0.0030406i at different scattering angles $\theta$ (indicated by color).

[Figure]

Figure S6: Comparison of IGOM (dotted lines) with TMM (solid lines) for **oblate spheroids with $\epsilon'=2.0$** and $m=1.52+0.0030406i$ at different scattering angles $\theta$ (indicated by color).

[Figure]

Figure S7: Comparison of IGOM (dotted lines) with TMM (solid lines) for **oblate spheroids with $\epsilon'=2.8$** and $m=1.52+0.0030406i$ at different scattering angles $\theta$ (indicated by color).

[Figure]

Figure S8: Comparison of IGOM (dotted lines) with TMM (solid lines) for **oblate spheroids with** $\epsilon'$**=5.0** and $m$=1.52+0.0030406i at different scattering angles $\theta$ (indicated by color).

**S4 Example for decomposition of a mode into contributions from data set grid points**

Suppose, a mode has spheroids with $m_r = 1.53$, $m_i = 0$, and $\epsilon_m = 1.92$. Then the MOPSMAP grid points of each dimension and their weights are

- $m_{r,i} = 1.52$ with $w_{m_r,i} = 0.75$

- $m_{r,i+1} = 1.56$ with $w_{m_r,i+1} = 0.25$

- $m_{i,j} = 0$ with $w_{m_i,j} = 1$

- $m_{i,j+1} = 0.0005375$ with $w_{m_i,j+1} = 0$

- $\epsilon_{m,k} = 1.8$ with $w_{\epsilon_m,k} = 0.4$

- $\epsilon_{m,k+1} = 2.0$ with $w_{\epsilon_m,k+1} = 0.6$

The contributions from the different $(m_r, m_i, \epsilon_m)$-grid points (i.e., netcdf files) included in the data set are therefore

1. $m_{r,i} = 1.52$, $m_{i,j} = 0$, $\epsilon_{m,k} = 1.8$ with $w = 0.75 \cdot 1 \cdot 0.4 = 0.30$

2. $m_{r,i} = 1.52$, $m_{i,j} = 0$, $\epsilon_{m,k+1} = 2.0$ with $w = 0.75 \cdot 1 \cdot 0.6 = 0.45$

3. $m_{r,i} = 1.52$, $m_{i,j+1} = 0.0005375$, $\epsilon_{m,k} = 1.8$ with $w = 0.75 \cdot 0 \cdot 0.4 = 0.00$

4. $m_{r,i} = 1.52$, $m_{i,j+1} = 0.0005375$, $\epsilon_{m,k+1} = 2.0$ with $w = 0.75 \cdot 0 \cdot 0.6 = 0.00$

5. $m_{r,i+1} = 1.56$, $m_{i,j} = 0$, $\epsilon_{m,k} = 1.8$ with $w = 0.25 \cdot 1 \cdot 0.4 = 0.10$

6. $m_{r,i+1} = 1.56$, $m_{i,j} = 0$, $\epsilon_{m,k+1} = 2.0$ with $w = 0.25 \cdot 1 \cdot 0.6 = 0.15$

7. $m_{r,i+1} = 1.56$, $m_{i,j+1} = 0.0005375$, $\epsilon_{m,k} = 1.8$ with $w = 0.25 \cdot 0 \cdot 0.4 = 0.00$

8. $m_{r,i+1} = 1.56$, $m_{i,j+1} = 0.0005375$, $\epsilon_{m,k+1} = 2.0$ with $w = 0.25 \cdot 0 \cdot 0.6 = 0.00$

The weighted sum of these 8 contributions (with 4 of them being zero) gives the optical properties of the mode.